

# A computational framework for processing time-series of earth observation data based on discrete convolution: global-scale historical Landsat cloud-free aggregates at 30 m spatial resolution

Davide Consoli[1], Leandro Parente[1], Rolf Simoes[1], Murat Şahin[1], Xuemeng Tian[1,2], Martijn Witjes[1,2], Lindsey Sloat[3] and Tomislav Hengl[1]

[1] OpenGeoHub Foundation, Doorwerth, Netherlands
[2] Laboratory of Geo-Information Science and Remote Sensing, Wageningen University and Research, Wageningen, Netherlands
[3] Land & Carbon Lab, World Resources Institute (WRI), Washington DC, United States

Corresponding author
Davide Consoli,
davide.consoli@opengeohub.org

## ABSTRACT

Processing large collections of earth observation (EO) time-series, often petabyte-sized, such as NASA's Landsat and ESA's Sentinel missions, can be computationally prohibitive and costly. Despite their name, even the Analysis Ready Data (ARD) versions of such collections can rarely be used as direct input for modeling because of cloud presence and/or prohibitive storage size. Existing solutions for readily using these data are not openly available, are poor in performance, or lack flexibility. Addressing this issue, we developed TSIRF (Time-Series Iteration-free Reconstruction Framework), a computational framework that can be used to apply diverse time-series processing tasks, such as temporal aggregation and time-series reconstruction by simply adjusting the convolution kernel. As the first large-scale application, TSIRF was employed to process the entire Global Land Analysis and Discovery (GLAD) ARD Landsat archive, producing a cloud-free bi-monthly aggregated product. This process, covering seven Landsat bands globally from 1997 to 2022, with more than two trillion pixels and for each one a time-series of 156 samples in the aggregated product, required approximately 28 hours of computation using 1248 Intel® Xeon® Gold 6248R CPUs. The quality of the result was assessed using a benchmark dataset derived from the aggregated product and comparing different imputation strategies. The resulting reconstructed images can be used as input for machine learning models or to map biophysical indices. To further limit the storage size the produced data was saved as 8-bit Cloud-Optimized GeoTIFFs (COG). With the hosting of about 20 TB per band/index for an entire 30 m resolution bi-monthly historical time-series distributed as open data, the product enables seamless, fast, and affordable access to the Landsat archive for environmental monitoring and analysis applications.

# INTRODUCTION

Raw earth observation (EO) data can rarely be used as a direct input to machine learning (ML) models or zonal statistics, as they often contain a significant amount of clouds, atmospheric features and artifacts. Several EO datasets, such as Landsat (*Woodcock et al., 2008*), Moderate Resolution Imaging Spectroradiometer (MODIS) (*Justice et al., 2002*) and Sentinel (*Spoto et al., 2012*) collections, can also be found as Analysis Ready Data (ARD) products (*Frantz, 2019*; *Potapov et al., 2020*), where several preprocessing steps are applied to provide more complete and consistent data, and more straightforward access to users. However, most of the currently available ARD products are affected by sensor artifacts, atmospheric contaminants (*e.g.*, clouds), and data gaps that jeopardize the use of the data set in practical applications, as discussed in recent publications on analysis-readiness of EO data (*Baumann, 2024*). Furthermore, when a long time-series of moderate and high resolution EO data is collected, the size of the dataset can exceed the petabyte (PB), making data storage, access, and elaboration prohibitive for most potential users (*Balsamo et al., 2018*). Solutions to these problems include data imputation (often named "*gap-filling*"), smoothing, outlier removal, space and/or time aggregation, decomposition, data-fusion and image compression. Several methods used for these processing tasks involve the discrete convolution between the input time-series and a convolution kernel that varies depending on the application. In general, for such methods, the discrete convolution also represents the largest computational effort for their application. This observation inspired us to develop a framework that we named Time-Series Iteration-free Reconstruction Framework (TSIRF) which embodies several processing tools for time-series. Within TSIRF, different methods can be applied by properly setting the convolution kernels. This allows to focus the development endeavor in optimizing a small number of operations and apply different methods by only modifying the convolution kernel.

Temporal aggregation of satellite time-series can allow reducing storage size while maintaining most useful information for common land cover classification applications (*Carrasco et al., 2019*). In addition, it also serves as partial data imputation for missing values in the time-series (*Carrasco et al., 2022*). However, since temporal aggregation is usually applied on large raw datasets, it is fundamental that its implementation is highly optimized. Therefore, TSIRF is a good framework for implementing such tools.

Cloud and artifact detection algorithms allows to mask and remove contaminated regions of images that could lead to biased results. However, the presence of such contaminants create data gaps in affected regions. Since several modeling techniques and

statistical analysis require the input data to be complete, *i.e.*, no missing values or data gaps, the application of imputation techniques is required (*Hermosilla et al., 2015*; *Radeloff et al., 2024*). In the context of time-series, those techniques are often defined time-series reconstruction methods, or more broadly gap-filling methods (we try to avoid the usage of this term since it can be misleading). TSIRF can be used to apply some iteration-free time-series reconstruction strategies, enabling its usage on large amount of pixels and/or long time-series.

Finally, because time-series often suffer from the presence of noise and outliers, smoothing techniques are often used to improve the quality of data (*Atkinson et al., 2012*; *Chen et al., 2004*). Some of these techniques are combined in sequence with simple data imputation strategies to improve their reconstruction performance. Notably, Savitzky–Golay (*Schafer, 2011*) and Lanczos (*Duchon, 1979*) filters, are also based on convolution between the original time-series and the specific convolution kernel. By properly setting the TSIRF convolution kernel it is possible to combine the time series reconstruction with the smoothing in one single operation, further reducing the computational costs.

Despite the abundance of methods for time series processing available in the literature, few of them are suitable for application on global-scale historical earth observation data, due to the prohibitive computational cost (*Siabi, Sanaeinejad & Ghahraman, 2020*). In addition, most of their implementations are not provided as open-source code, and/or, in the case of methods that require it, the external data are not openly available. This is the research gap that we wanted to fill by presenting TSIRF. Furthermore, as a first large-scale application of TSIRF, this work also presents a bimonthly aggregated, cloud-free, analysis-ready, and cloud-optimized (ARCO) Landsat collection based on the Global Land Analysis and Discovery (GLAD) Landsat ARD (version 2). Notably, *Moreno-Martínez et al. (2020)* proposes a data-fusion approach that uses Landsat and MODIS images to generate a product with similar target applications and characteristics. However, our works differ in the source of data, since it is based on the GLAD Landsat ARD product instead of the original United States Geological Survey (USGS) images, and in the methodology used for the data processing. Additionally, we developed our open-source code using Python combined with C++ and produced the data using our own computing infrastructure instead of the Google Earth Engine (GEE) platform. Evidently, this brings advantages and disadvantages in terms of accessibility of the data that are discussed in the following. Other works used Synthetic Aperture Radar (SAR) images to reconstruct surface reflectance one, taking advantage of the penetrability of clouds and atmospheric contaminants of SAR satellites. For instance, *Hamelberg (2020)* and *Pipia et al. (2019)* used Sentinel-1 data to reconstruct Sentinel-2 data. However, the earliest available SAR images with spatial and temporal resolution comparable with Landsat are indeed the ones produced by Sentinel-1. Since the mission was launched only in 2014, at the best of our knowledge, there are no valid alternatives that can be used to apply this strategy on previous years.

We first introduce the theoretical background and implementation details of TSIRF. In addition, we include some guidelines for its usage in different time-series processing tasks. We then report the results obtained for the performance comparison of discrete convolution computational backends and of some selected time-series reconstruction methods, together with the analysis of the produced Landsat-bimonthly aggregated dataset. Finally, we provide a deeper analysis of the results obtained and discuss future development directions and implications of our findings. In this article, we specifically try to answer the following research questions:

**RQ1** Is it possible to encompass several EO time-series processing method in a single framework?

**RQ2** Is it possible to optimize the performance time-series processing tools in order to apply them to PetaByte size datasets with reduced computational resources?

**RQ3** Which is the best-performing highly-scalable time-series reconstruction method for global-scale historical EO data?

**RQ4** Which strategy can be used to produce a time-series of global-scale historical Landsat data compact in storage size and free of data gaps?

The data reported in this work are available under an open data license from:

- https://stac.openlandmap.org/landsat_glad.swa.ard2_bimonthly/collection.json;
- https://stac.openlandmap.org/landsat_glad.swa.ard2_yearly.p50/collection.json;
- https://github.com/openlandmap/scikit-map/blob/feat_tsirf/CATALOG.md;
- https://zenodo.org/records/11150343.

The code used to produce analysis and visualizations is available at https://github.com/openlandmap/scikit-map/tree/feat_tsirf. Portions of this text were previously published as part of a preprint (*Consoli et al., 2024*).

# MATERIALS AND METHODS

## A computational framework for time-series processing

The implementation of several tools used to preprocess Earth Observation (EO) time-series involves the usage of a discrete convolution. Generally speaking, convolution is the mathematical operation that takes two functions, slides them one over the other, multiply them point by point, and integrates their product. In fact, the methods that involve a *sliding window* in time (or also in space) are often based on convolution, and many of them are used for EO data processing. Combining the convolution operation with the Hadamard (element-wise) product and division, TSIRF can also be used to perform time-series reconstruction and temporal data aggregation. In addition, the implementation of the framework is provided using three different computational back-ends. The selection of the most convenient back-end depends on the application and used computational infrastructure. Implementation details are provided after the definitions and notation.

**Table 1 Summary of defined symbols.**

| Symbol | Description |
| --- | --- |
| $T_s$ | Sampling period |
| $N_s$ | Number of samples of the time series |
| $N_p$ | Number of elements of the convolution kernel in the relative past |
| $N_f$ | Number of elements of the convolution kernel in the relative future |
| $N_w$ | Total number of elements of the convolution kernel |
| $N_e$ | Number of the periodic extension of the time-series |
| $\boldsymbol{w}$ | Vector containing the convolution kernel samples |
| $\boldsymbol{w_p}$ | Sub-vector of $\boldsymbol{w}$ with elements from the relative past |
| $\boldsymbol{w_f}$ | Sub-vector of $\boldsymbol{w}$ with elements from the relative future |
| $N_r$ | Number of time-series to which a the method is applied |
| $N_a$ | Aggregation factor for temporal aggregation |
| $\varepsilon$ | Floating point machine precision |

## Theoretical background and notation

Consider two real-valued functions $v(t)$ and $w(t)$, where $v(t)$ represents the variable of interest in function of time (*e.g.*, blue band value for a certain pixel in EO time-series), and $w(t)$ is the convolution kernel, input that is defined depending on the target processing task. The uni-dimensional convolution between the two variables is defined as

$$(v * w)(t) = \int_{-\infty}^{\infty} v(\tau)w(t - \tau)d\tau,$$

where the integrated variable $\tau$ is used to slide one of the function on the other one (*Oppenheim & Schafer, 1975*). The time-discrete version of convolution can be defined for uniformly sampled versions of the two variables as

$$(v * w)[n] = \sum_{m=-\infty}^{\infty} v[m]w[n - m],$$

where $n$ is the index of a time sample, with sampling period $T_s$ and frequency $f = \frac{1}{T_s}$. As detailed in the following, in order to optimize computational performance in case of long time-series, it is convenient to perform the discrete convolution using Fast Fourier Transform (FFT). In order to do that, we extend here the notation to the also define the circular convolution. Noting that the observed variable $v[n]$ is only available for a limited number of samples $N_s$, relative to the observation time-frame $(T_0, T_0 + N_s T_s)$, and supposing to have a convolution kernel with limited support in the sample range $(-N_p, N_f)$, where $N_p$ is the number of non zero samples in the "relative past" and $N_f$ is the number of non zero samples in the "relative future", it is possible to define a periodic extensions of the two time-discrete variables, $v_e[n]$ and $w_e[n]$, with periodicity $N_e T_s$ where

$$N_e = N_s + max(N_p, N_f).$$
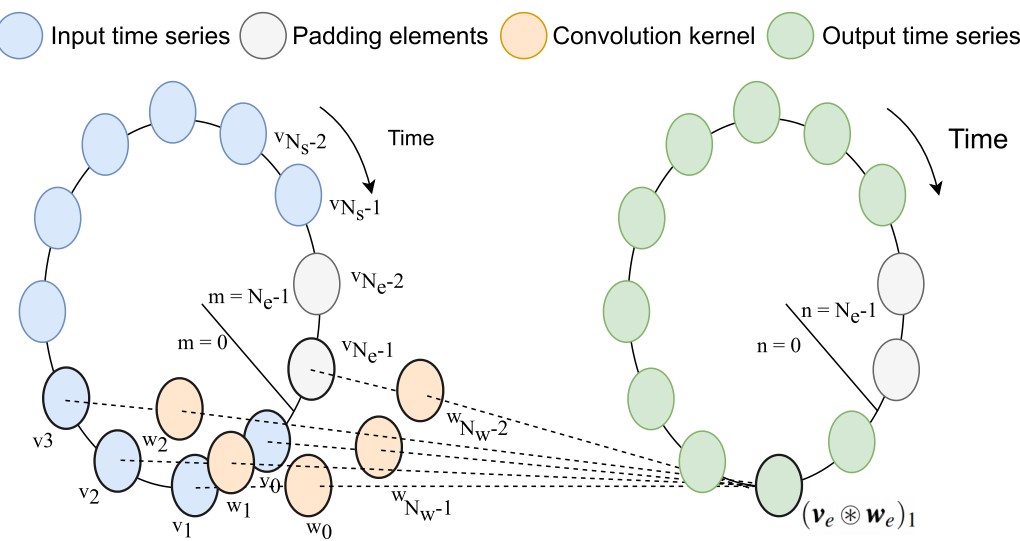

**Figure 1 Graphical representation of circular convolution.** In this example, the number of samples in the input time-series stored in $v$ is $N_s = 10$, while the convolution kernel stored $w$ has $N_p = N_f = 2$ weights for in the relative past and relative future (for $N_w = 5$ in total including the relative present). After padding, this lead to $N_e = 12$ as size of the extended version of the vectors. Each element of the output vector $v_e \circledast w_e$ is computed as the summation between the products of the sliding kernel vector and the input time-series.

In order to facilitate the reading of the equations, the symbology is summarized in Table 1. Following this notation, we define their circular convolution (*Bamieh, 2018*) as

$$(v_e \circledast w_e)[n] = \sum_{m=0}^{N_e-1} v_e[m] w_e[n-m]. \tag{1}$$

Note that from previously defined range of the support of $w[n]$, $(-N_p, N_f)$, the total number of element in $w[n]$ support is $N_w = N_p + N_f + 1$. Assuming that $N_p < N_s > N_f$, periodic extension of $v_e[n]$ and $w_e[n]$ can be performed with zero-padding for newly introduced elements such that $(v * w)[n] = (v_e \circledast w_e)[n]$. A visual interpretation of the circulant convolution operation, inspired by the one in *Guo et al. (2023)*, is shown in Fig. 1.

In order to describe the different computational backends for the convolution, we introduce the vectorial notation of the previously defined quantities. The values of the time-discrete variables $v[n]$ and $w[n]$ are respectively stored in the row vectors.

$$v = \begin{bmatrix} v_0 \\ v_1 \\ \vdots \\ v_{N_s-2} \\ v_{N_s-1} \end{bmatrix}^\top = \begin{bmatrix} v[0] \\ v[1] \\ \vdots \\ v[N_s-2] \\ v[N_s-1] \end{bmatrix}^\top \quad and \quad w = \begin{bmatrix} w_0 \\ w_1 \\ \vdots \\ w_{N_f-1} \\ w_{N_w-N_p} \\ \vdots \\ w_{N_w-2} \\ w_{N_w-1} \end{bmatrix}^\top = \begin{bmatrix} w[0] \\ w[1] \\ \vdots \\ w[N_f-1] \\ w[-(N_p-1)] \\ \vdots \\ w[-2] \\ w[-1] \end{bmatrix}^\top .$$

Note that the indexing notation concatenates the values of the relative past in the convolution kernel after those of the relative future, from the least recent, $w[-(N_p - 1)]$, to the most recent, $w[-1]$ (Fig. 1). Similarly, the vectors storing their periodic extensions are defined as

$$
\boldsymbol{v_e} = \begin{bmatrix} v_{e;0} \\ v_{e;1} \\ \vdots \\ v_{e;N_s-1} \\ v_{e;N_e-max(N_p,N_f)} \\ \vdots \\ v_{e;N_e-2} \\ v_{e;N_e-1} \end{bmatrix}^\top = \begin{bmatrix} v[0] \\ v[1] \\ \vdots \\ v[N_s-1] \\ 0 \\ \vdots \\ 0 \\ 0 \end{bmatrix}^\top , \quad \boldsymbol{w_e} = \begin{bmatrix} w_{e;0} \\ w_{e;1} \\ \vdots \\ w_{e;N_f-1} \\ w_{e;N_f} \\ \vdots \\ w_{e;N_f-N_p-1} \\ w_{e;N_e-N_p} \\ \vdots \\ w_{e;N_e-2} \\ w_{e;N_e-1} \end{bmatrix}^\top = \begin{bmatrix} w[0] \\ w[1] \\ \vdots \\ w[N_f-1] \\ 0 \\ \vdots \\ 0 \\ w[-(N_p-1)] \\ \vdots \\ w[-2] \\ w[-1] \end{bmatrix}^\top .
$$

As highlighted in the notation, the (zero) padding elements are inserted at the end of the input time-series and in the center of the convolution kernel, in order to correctly align the two vectors.

Following this indexing convention, the convolution between $v[n]$ and $w[n]$ can be computed as

$$
(v_e \circledast w_e)[n] = \sum_{m=0}^{N_e-1} v_m w_{e;(N_e+m-n)\%N_e}. \tag{2}
$$

We now define the vector with elements containing the result of the convolution and of the circulant convolution respectively as

$$
\boldsymbol{v * w} = \begin{bmatrix} (v*w)_0 \\ (v*w)_1 \\ \vdots \\ (v*w)_{N_s-2} \\ (v*w)_{N_s-1} \end{bmatrix}^\top = \begin{bmatrix} (v*w)[0] \\ (v*w)[1] \\ \vdots \\ (v*w)[N_s-2] \\ (v*w)[N_s-1] \end{bmatrix}^\top , \quad \boldsymbol{v_e \circledast w_e} = \begin{bmatrix} (v_e \circledast w_e)_0 \\ (v_e \circledast w_e)_1 \\ \vdots \\ (v_e \circledast w_e)_{N_e-2} \\ (v_e \circledast w_e)_{N_e-1} \end{bmatrix}^\top = \begin{bmatrix} (v_e \circledast w_e)[0] \\ (v_e \circledast w_e)[1] \\ \vdots \\ (v_e \circledast w_e)[N_e-2] \\ (v_e \circledast w_e)[N_e-1] \end{bmatrix}^\top .
$$

From these definitions, we have that the elements of vector $v * w$ coincide with the first $N_s$ elements of $v_e \circledast w_e$. Finally, we introduce the circulant matrix with columns derived by rolling the vector $w_e$ as

$$
\boldsymbol{W_e} = \begin{bmatrix} w_{0,0} & w_{0,1} & \cdots & w_{0,N_e-2} & w_{0,N_e-1} \\ w_{1,0} & w_{1,1} & \cdots & w_{1,N_e-2} & w_{1,N_e-1} \\ \vdots & \vdots & \ddots & \vdots & \vdots \\ w_{N_e-2,0} & w_{N_e-2,1} & \cdots & w_{N_e-2,N_e-2} & w_{N_e-2,N_e-1} \\ w_{N_e-1,0} & w_{N_e-1,1} & \cdots & w_{N_e-1,N_e-2} & w_{N_e-1,N_e-1} \end{bmatrix} = \begin{bmatrix} w_{e;0} & w_{e;N_e-1} & \cdots & w_{e;2} & w_{e;1} \\ w_{e;1} & w_{e;0} & \cdots & w_{e;3} & w_{e;2} \\ \vdots & \vdots & \ddots & \vdots & \vdots \\ w_{e;N_e-2} & w_{e;N_e-3} & \cdots & w_{e;0} & w_{e;N_e-1} \\ w_{e;N_e-1} & w_{e;N_e-2} & \cdots & w_{e;1} & w_{e;0} \end{bmatrix} .
$$

The same indexing notation is used for all the other matrices. Clearly, the circulant matrix $W_e$ can be used to compute the elements of $v_e \circledast w_e$ performing the vector-matrix product

$$v_e \circledast w_e = v_e W_e.$$

From the top left quarter of the matrix $W_e$, we also define the sub-matrix

$$W = \begin{bmatrix} w_{e;0} & w_{e;N_e-1} & \cdots & w_{e;N_s+1} & w_{e;N_s} \\ \underline{w_{e;1}} & w_{e;0} & \cdots & w_{e;N_s+2} & w_{e;N_s+1} \\ \vdots & \vdots & \ddots & \vdots & \vdots \\ \underline{w_{e;N_s-2}} & \underline{w_{e;N_s-3}} & \cdots & w_{e;0} & w_{e;N_e-1} \\ \underline{w_{e;N_s-1}} & \underline{w_{e;N_s-2}} & \cdots & \underline{w_{e;1}} & w_{e;0} \end{bmatrix}$$

where elements of the convolution kernel associated with the *relative past*, below the main diagonal, are under-lined, while the ones associated with the *relative future*, above the main diagonal, are upper-lined to emphasize the interpretability of the defined operation. In the case of zero-padding, this results in

$$v \circledast w = vW. \tag{3}$$

Note that matrices $W_e$ and $W$, respectively, are of circulant and Toeplitz type. It is well known that matrix-vector products involving circulant or Toeplitz matrices can be efficiently computed using FFT (*Strang, 1986*). For several applications, this represents a convenient solution in terms of computational cost and time, as shown in the results section.

Targeting large-scale applications, we consider the case in which the same processing method is applied to multiple time-series. We so define the matrix $V$, which contains on each row one of the $N_r$ input time-series. Clearly, the matrix resulting from the matrix-matrix product $VW$ will contain on each row the convolution between the corresponding time-series and the convolution kernel $w[n]$. However, the same result can be obtained by explicit iterative summation, as in Eq. (1), or as detailed in the following sections, using FFT. Finally, in order to deal with missing values in the original time-series, we define the matrix

$$M = \begin{bmatrix} m_{0,0} & m_{0,1} & \cdots & m_{0,N_s-2} & m_{0,N_s-1} \\ m_{1,0} & m_{1,1} & \cdots & m_{1,N_s-2} & m_{1,N_s-1} \\ \vdots & \vdots & \ddots & \vdots & \vdots \\ m_{N_r-2,0} & m_{N_r-2,1} & \cdots & m_{N_r-2,N_s-2} & m_{N_r-2,N_s-1} \\ m_{N_r-1,0} & m_{N_r-1,1} & \cdots & m_{N_r-1,N_s-2} & m_{N_r-1,N_s-1,} \end{bmatrix}$$

containing the validity-mask of the matrix $V$, with elements value defined as

$$m_{i,j} = \begin{cases} 1 & \text{if } v_{i,j} \text{ is a valid sample;} \\ 0 & \text{otherwise.} \end{cases}$$

### The processing framework

We now introduce the definition of TSIRF. We assume that the elements of $V$ that contain missing values, so where the elements of $M$ are equal to 1, are set to 0. With this setting, the

matrix $M$ can be used to normalize the result of the discrete convolution depending on the availability of the sample. In particular, we apply SIRLCE as

$$\widetilde{V} = (VW) \oslash (MW), \tag{4}$$

where $\oslash$ denotes the Hadamard (element-wise) division. This matrix equation with the above definitions allows for instance to efficiently compute weighted averages with sliding window weights between available samples, and is the base of the time-series reconstruction method presented in the following. Equivalently, the computation of each element of $\widetilde{V}$ reads like

$$\widetilde{v}_{i,j} = \frac{\sum_{k=0}^{N_s-1} v_{i,k} w_{k,j}}{\sum_{k=0}^{N_s-1} m_{i,k} w_{k,j}}.$$

Finally, we want to highlight that it is also possible to includes arbitrary element-wise scaling of available samples in the original time-series by performing the Hadamard (element-wise) product between the desired scaling matrix and both $V$ and $M$ before applying Eq. (4).

### Efficient computation of circular convolutions

Different strategies can be used to compute the discrete convolution between $w[t]$ and the time-series stored in $V$ (and in $M$). The current implementation included in the Python library *scikit-map* includes three different backed implementations for its computation. The first method, in the following text referred to as *Summation*, consists of simply computing the elements of the final result by summation as in Eq. (2). However, only the non-zero elements of the convolution kernel contribute to the result of the operation, so only $N_w$ iterations of the summation are necessary. Indeed, as proved by the benchmarks, this is particularly effective when the convolution kernel has few non-zero elements compared to the length of the time-series (*e.g.*, Savitzky–Golay filters).

The second method explicitly computes the matrix-matrix products present in Eq. (4). For this reason, this backend is referred to as *Matrix* in the following. This strategy is generally convenient when the same processing strategy is applied to several time-series, so when $N_r$ is high (*e.g.*, time-series reconstruction of high spatial resolution EO data). This comes from the higher efficiency of the CPU caching and, possibly, by the lower asymptotic complexity of the linear algebra library implementation of the matrix-matrix product.

Finally, as mentioned before, it is possible to compute the convolution using the properties of the Fourier series. It is well known that the eigenvectors of any circulant matrix coincide with the columns of the discrete Fourier transform (DFT) matrix $F$ (*Flannery et al., 1992*; *Gray, 2006*), sampled version of the harmonic functions (sinusoidal waves). In fact, the matrix $W_e$ can be diagonalized multiplying it by $F$, leading to the decomposition

$$W_e = F \Lambda_{W_e} F^{-1}, \tag{5}$$

where $\Lambda_{W_e}$ is the diagonal matrix containing the eigenvalues of $W_e$, in spectral order. In addition, the eigenvalues of $W_e$ can be computed using only one of its columns

$$\lambda_{W_e} = w_e F, \tag{6}$$

where $\lambda_{W_e}$ coincide with the main diagonal of $\Lambda_{W_e}$. Substituting Eq. (5), and then Eq. (6), in Eq. (3), we have

$$v_e \circledast w_e = v_e F \Lambda_{W_e} F^{-1} = ((v_e F) \odot (w_e F)) F^{-1},$$

where $\odot$ is the Hadamard (element-wise) product. Indeed, in the Fourier domain the convolution operation is translated in the multiplication, and *vice-versa*. Using the fast implementation of the 1D DFT (the 1D FFT), the matrix-vector multiplication between $F$ and a vector can be obtained applying the forward FFT transform, while one involving its inverse, using the backward or inverse fast Fourier transform (IFFT). Using this approach, the computational complexity of the entire operation scales asymptotically as $\mathcal{O}(N_e \log_2 N_e)$, reducing it from quadratic to quasi-linear. This makes the *FFT* backend a valuable solution in the case of very long time-series (or high temporal resolution). A deeper analysis of the computational complexity of the backends is provided in the discussion section.

### Implementation details

Some care must be taken when implementing equations Eq. (4). In particular, when elements in the resultant of the convolution operation involving the validity mask $M$ are expected to be exact zeros, it means that no values in the original time-series were available in the corresponding temporal range, and should remain data gaps. However, numerical noise, due to round-off error and numerical cancellation, could slightly modify the result of the convolution, up to the numerical precision of the machine (*e.g.*, $10^{-16}$ for floats in double precision) (*Goldberg, 1991*). If a numerical error makes such elements different from exact zeros, the remaining gaps will not be detected, and numerical noise will be used as actual data.

To overcome this issue, the first step is to define a convolution kernel where the smallest non-zero element is at least one or two orders of magnitudes larger than the numerical precision (to keep some margin). Then, once the convolution between the validity mask and the convolution kernel is performed, the resulting values that are inferior to the smallest nonzero element in the convolution kernel can be clipped to zero, since any non-zero element should be larger or equal to that because the mask is composed by either one or zeros. This is the default behavior implemented in the *scikit-map* library. Note that, however, in the case where also the element-wise scaling is applied, the masking threshold should not be set to the minimum value of the convolution kernel, but to the product between it and the maximum value of the matrix $S$. This imply that the additional margin should be kept when setting the smallest non-zero value in the convolution kernel.

Finally, for a more convenient usage, the library interface was implemented defining three input parameter: $w_0$, $w_p$ and $w_f$. The first parameter is a scalar defining

the value of $w[0]$, the while the second and the third are vectors containing, in chronological order, the elements of $w$ for the relative past and the relative future, respectively.

## Guidelines for convolution kernel definition

As anticipated, different time-series processing tasks can be accomplished by properly defining the convolution kernel in TSIRF. This section provides some details on how to set the convolution kernel based on the target application. All convolution kernel elements are supposed to be in the range $\varepsilon < w[n] \leq 1$, where $\varepsilon$ is the machine accuracy previously mentioned (*e.g.*, for double float $\varepsilon = 2.22 \cdot 10^{-16}$). Using an higher upper limit instead of 1 would have no impact on the final result, but it is a preferable choice to avoid potential overflows. For all the following application the value of $w[0]$ is set to 1 and, for sake of brevity, this will not repeated. As a general note, in case the application requires to respect the causality principle, it is necessary to do not use nonzero weights for the values associated with the *relative future* of the convolution kernel. A property of TSIRF is to be an interpolant, so the reconstructed values will be numerically bounded between the minimum and the maximum of the available ones.

As first application, we describe how to perform temporal aggregation by averaging a sliding window of images. If we want an aggregation factor of $N_a$ simply by setting $N_a$ consecutive elements of the convolution kernel to 1. For instance, these elements could be centered in $w[0]$ (included in the 1 element count). However, after the application of TSIRF, the output $\widetilde{V}$ would still have $N_s$ columns as input data. To obtain the aggregated time-series it is necessary to keep one column every $N_a$. More conveniently, when using the *Matrix* backend, it is possible to remove the unused columns directly from $W$ before performing the matrix-matrix, in order to save unnecessary computation. A graphical example and summary is reported in the first row of Table 2.

We now describe the implementation different methods for time series reconstruction. Some of the propose methods do not exactly match their standard counterpart but are approximations of them. For instance, piecewise linear interpolation, a popular choice for simple time-series reconstruction (*Potapov et al., 2021*; *Yin et al., 2017*), can be approximated applying TSIRF with a convolution kernel described by the function

$$w(t) = 1 - \left| \frac{t}{N_s T_s} \right| + \varepsilon \tag{7}$$

and represented in the second row of Table 2. The operation is indeed interpolant and linear, however not piecewise as usually implemented in literature. Compared to the standard piecewise version, this implementation does not require to iterate over the time-series to fine the two nearest available values. The vectorization of the operation generally reduce the computational cost and as shown in the results section also lead to improved reconstruction performance compared to the standard version.

Another widely used strategy is the most recently available value propagation to reconstruct missing values. Similarly to the linear interpolation, this method can be approximated with TSIRF. In particular, using the convolution kernel

**Table 2 Summary and graphical representation of the convolution kernel setup guidelines for different applications.**

| Method | Convolution kernel shape | Description |
|---|---|---|
| Temporal aggregation by averaging |  | Set to 1 a number of consecutive elements of the convolution kernel equivalent to the aggregation factor $N_a$. Only keep one column out of $N_a$ in $\widetilde{V}$ (or more conveniently of $W$). |
| Approximated linear interpolation |  | Using the convolution kernel described by Eq. (7) results in a linear interpolant of the available values, with properties similar to the standard piecewise linear interpolation. |
| Approximated most recent image propagation |  | Using the convolution kernel described by Eq. (8), when the machine precision is not too coarse nor $N_s$ too high, results in a numerical approximation of the most recently available value propagation. |
| Seasonally weighted average (linear) |  | The function is described by Eq. (9) and can be used to reconstruct time series with known seasonality of period $T_{seas}$ (in this case one year). |
| Seasonally weighted average (exponential) |  | Similar to the previous row, but also including an envelope attenuation to reduce land-cover changes propagation. The function is described by Eq. (10). |

$$w(t) = 10^{\left|\frac{t}{N_s T_s}\right| log_{10} \varepsilon}, \tag{8}$$

the weight associated with each available image decay exponentially with relative time distance, as shown in the third column of Table 2. If the numerical precision is not too coarse nor the number of samples too high, each weight can differ of order of magnitudes, propagating numerically the most recent available image.

An additional strategy for EO time-series reconstruction implemented within the TSIRF framework relies on the prior that most of the global land surface reflectance shows an yearly seasonal pattern (typically with one year periodicity). In particular, a higher weight is given to images with relative time distance nearer to integer multiples of one season period $T_{seas}$. From this comes its name Seasonally Weighted Average (SWA). An example of such a convolution kernel is represented in the forth row of Table 2. This can be implemented using a triangular wave function of period $T_{seas}$, that reads like

$$w(t) = 1 - 2\left|\frac{t}{T_{seas}} - \left\lfloor \frac{t}{T_{seas}} + \frac{1}{2} \right\rfloor\right| + \varepsilon, \tag{9}$$

where $\lfloor \rfloor$ is the floor function.

Furthermore, to avoid potential propagation land-cover change along the time-series, it is preferable to also give higher priority to images that are more recent. This is done by subtracting a scaled modulo function from Eq. (9). To enhance the numerical impact of the weighting, like in the case of most recent image propagation, the function is used as an exponent of 10. Two input parameters are introduced to define the seasonal attenuation $A_{seas}$, and the envelope attenuation $A_{env}$, with the latter used to limit land-cover changes propagation. Both attenuations are interpreted in decibels (dB), leading to the final version of the weighting function

$$w(t) = 10^{-2\frac{A_{seas}}{10}\left|\frac{t}{T_{seas}} - \left\lfloor \frac{t}{T_{seas}} + \frac{1}{2} \right\rfloor\right| - \frac{A_{env}}{10}\left|\frac{t}{N_s T_s}\right|}. \tag{10}$$

Note that in consideration of the numerical problems previously described, it is recommendable to respect the condition $log_{10}\varepsilon < -\left(\frac{A_{seas}}{10} + \frac{A_{env}}{10}\right)$. A graphical example of the convolution kernel is represented in the bottom row of Table 2, while a detailed interpretation of the method is shown in Fig. 2. On top-left the input time-series with missing samples that need to be reconstructed. In the bottom left of the scheme, the graph shows the weights of the convolution kernel $w[t]$ associated with $w_e$. In the central column, for each of the three missing values, the convolution kernel is shifted and overlapped with the original time series to highlight the method functioning. The right most column we find the reconstructed time-series.

## Analysis ready and cloud-free Landsat bimonthly aggregates

The Landsat ARD provided by the GLAD team at the University of Maryland is one of the few globally consistent archives for historical time-series of normalized surface reflectance harmonizing the entire Landsat satellite collections (*Potapov et al., 2020*). However, the raw GLAD Landsat ARD images include cloud contamination and image artifacts that could propagate in derived product if used as direct input for modeling land-use (LU) or

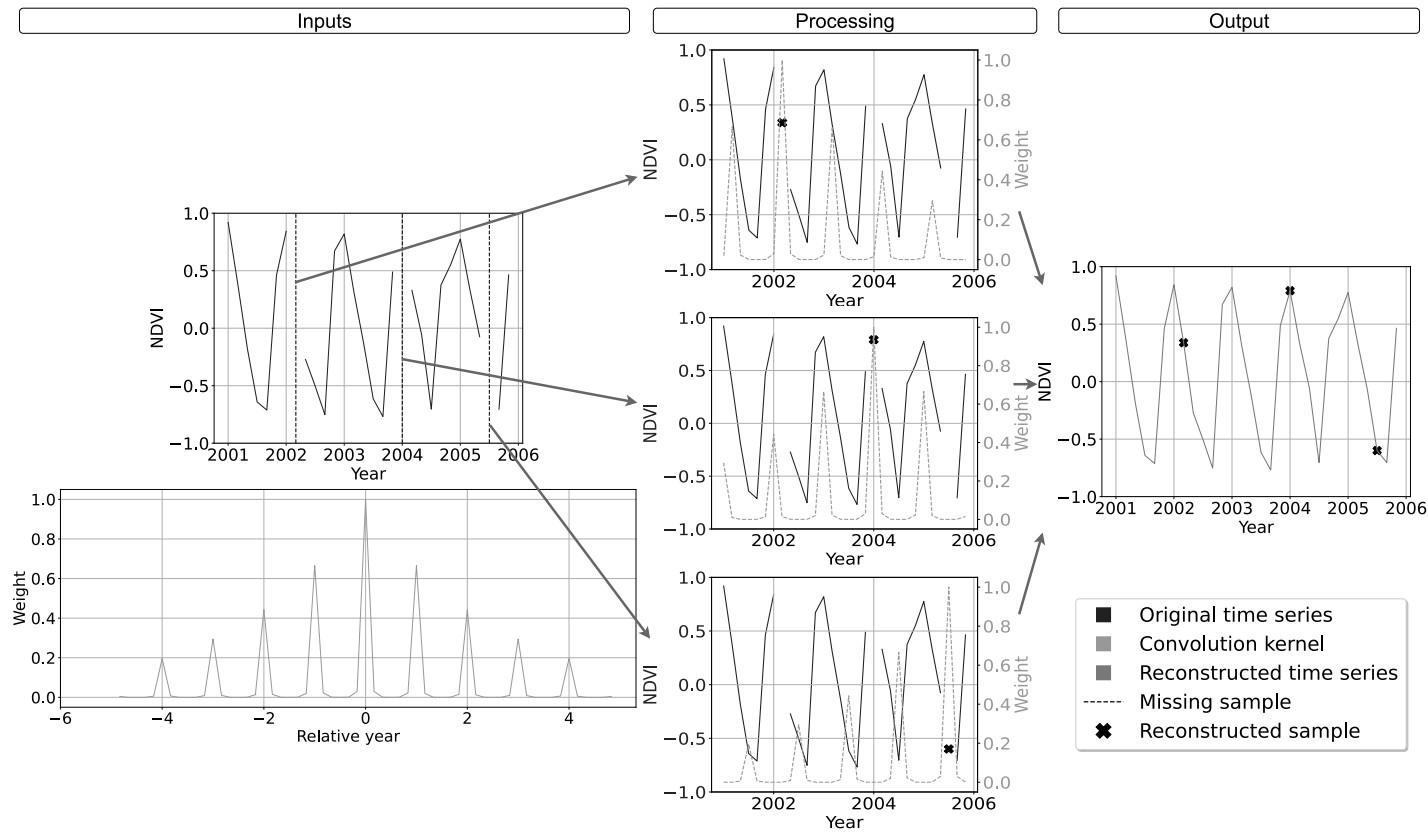

**Figure 2 Schematic description of SWA functioning for time-series reconstruction.** On top-left the input time-series with missing samples that need to be reconstructed. On bottom-left the convolution kernel used as weight for the reconstruction, with higher weight for samples with same seasonality and lower weights for samples farther in time. In the central column, the reconstruction of each missing sample is performed centering on it the convolution kernel, summing the weighted available samples and renormalizing the result by the used weights. On the right the reconstructed time-series.            

land-cover (LC). In addition, the storage size of the whole collection amounts to about 1.4 PB, a size that can make prohibitive its usage due to download time and required hardware. For these reasons, we produced a derived collection removing contaminated part of images, performing weighted temporal aggregation, and applying SWA for reconstructing the times-series associated with each pixel. The so-produced images can be used as direct input for modeling or to produce global time-series of biophysical indices. Stored as cloud optimized GeoTIFF (COG) on S3 systems, they result in an ARCO product that can also be straightforwardly used for geo-spatial analysis. The production pipeline, schematized in Fig. 3 and described in the following, represents a first large-scale application of TSIRF.

The input dataset GLAD Landsat ARD is a tiled multi-band 16-days interval composite of images with 23 images per year from 1997 to 2022, for a total of 598 GeoTIFF images per tile saved as unsigned 16-bit integer, and a total storage size of about 1.4 PB (*Potapov et al., 2020*). The GLAD Landsat ARD harmonizes all available images from the different Landsat satellites (Landsat 5 TM, Landsat 7 ETM+ and Landsat 8 OLI/TIRS) from 1997 to present, providing one of the best sources of historical 30 m resolution surface reflectance

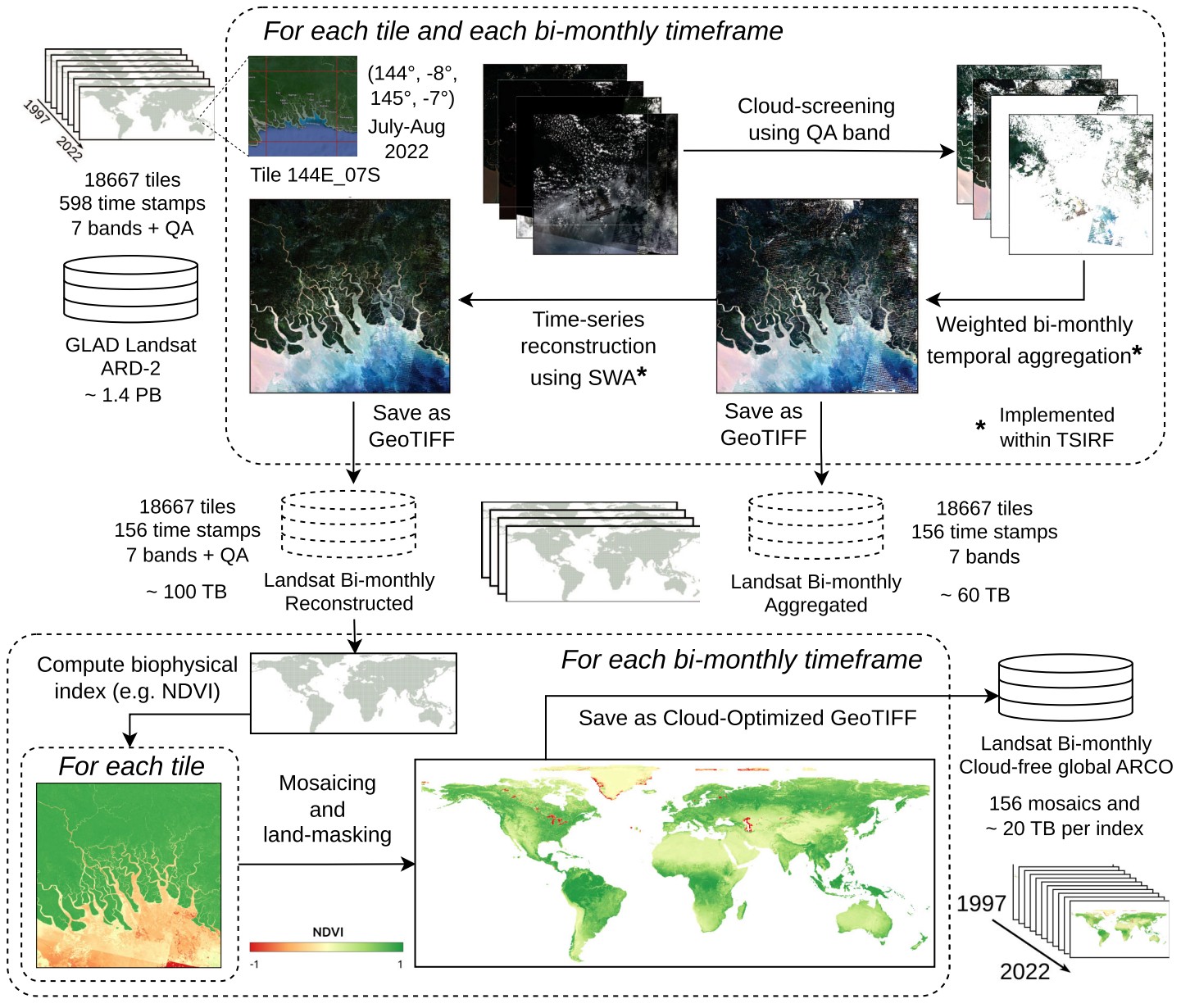

**Figure 3  Block scheme of Landsat ARD processing based on TSIRF.** In top left the input tiled dataset (seven bands + quaintly assessment, 30 m spatial resolution and 16-days time resolution). For each tile the whole time-series is sequence (i) cloud screened, (ii) time aggregated in bimonthly frames and (iii) reconstructed using SWA. Time aggregation and SWA are implemented within the TSIRF framework, and both their result are saved in a S3 storage system. The Landsat bimonthly Reconstructed dataset is used as input to compute biophysical indices, like the normalized difference vegetation index (NDVI), land-masked and stored as global mosaiced and cloud optimize GeoTIFFs (COG) in a S3 storage system. Base map © Google Hybrid.                           

time-series. In addition to the seven reflectance bands, blue, green, red, near infrared (NIR), short-wave infrared 1 (SWIR1) and short-wave infrared 2 (SWIR2) and the thermal band, the GLAD Landsat ARD includes a detailed quality flag (QA) that classifies each pixel as: land, water, cloud, cloud shadow, topographic shadow, hill shade, snow, haze, cloud proximity, shadow proximity, other shadows or buffered proximity of the previously mentioned ones. From this flag a space-time *not clear-sky* mask was derived, including the

points that are classified as cloud, cloud shadow, haze, cloud proximity, shadow proximity or other shadows. Those points are considered as gaps and are imputed with the following procedure.

From the 16-days time interval time-series, a bimonthly product was derived by performing weighted temporal aggregation. The aggregation was obtained using four images from the original product to produce one image in the aggregated product. Since each year has 23 images, the last image of the year, associated to the November-December time-frame, was produced aggregating the last three images of the relative year and the first images of the following year of the GLAD Landsat ARD (excluding November–December for the year 2022 for which only three images were used). To minimize cloud-induced artifacts in the time-series, a heuristic approach was employed, prioritizing images with lower cloud cover. For each image in the bi-month time-frame, a clear-sky fraction of the relative image was computed (*e.g.*, in an image where one-fourth of the pixels are considered as gap the clear-sky fraction is 0.75). The aggregation was then performed as a weighted average between the available values, where each pixel was weighted by the clear-sky fraction of the associated image. If no valid observation was found, the pixel was considered as "no-data" value in the output. The computation was performed using the previously described temporal aggregation of TSIRF with $N_a = 4$ and element-wise scaling of the values based on the clear-sky fraction. Finally, each time-frame/tile/band combination was saved in a separate GeoTIFF as unsigned 8-bit (byte) integer, with values ranging from 0 to 250 and associating the no-data values to 255. After compressing per chunks of 1,024 by 1,024 pixels with Deflate (*Deutsch, 1996*), the total storage size resulted in about 60 TB.

However, the aggregated product still contains several missing values in the images time-series. In particular, this is due to: (i) no available images during winter months in northern areas, (ii) cloud cover during the rain season in tropical areas, and (iii) a high revisit period during years in which fewer Landsat satellites are active, in particular when only Landsat 7 producing images affected by striping due to sensor malfunctioning was active (*Hermosilla et al., 2015*). To produce complete time-series, SWA was applied on top of the bimonthly aggregated product, for each band and each pixel. In particular, the $A_{seas}$ and $A_{env}$ parameters were optimized by grid-search optimization on the benchmark dataset (*Consoli et al., 2023*), consisting of 16-days time-series of NDVI images derived from the harmonized Landsat–Sentinel (HLS) collection from 2015 to 2022. The $A_{env}$ parameter was scaled to be consistent with the different time range of the two products. As a result of the optimization, we used $A_{seas} = 45\ dB$ and $A_{env} = 46\ dB$. To avoid reconstructions of missing values with future images, $\boldsymbol{w}_f$ was left empty. In addition to respect causality, this strategy allows one to consistently update the product when images when more recent ones are available applying the time-series reconstruction only on them, without modifying the previous images. However, this implies that missing values could still be present in case no images are available at the beginning of the time-series. Since this generally impacts only few regions and for a few time frames, the choice on how to perform imputation is left to the final user. For instance, it is possible to apply SWA again using images from the future only for the imputation of the remaining missing values. In

addition to the seven Landsat reconstructed bands, we also produced a clear sky flag specifying if the value was reconstructed or already available in the aggregated product. In case the value was already available in the aggregated product the flag is set to 250, while 255 represent again the no-data values. All other values of the flag are associated with reconstructed pixels. Also in this case, each time-frame/tile/band combination was saved and compressed, as for the aggregated product, leading to a total storage size resulting in about 100 TB of data.

As shown in the bottom of Fig. 3, the Landsat bimonthly reconstructed images can be used as input for LU/LC modeling, or alternatively, to produce remote-sensed biophysical indices. This was the case for the soil health monitoring data produced in *Tian et al. (2024)*. The production pipeline of each index used the reconstructed images to calculate the index values. Then each bimonthly time frame is landmasked to remove sea water pixel, and finally aggregated in global (or regional) mosaics saved as cloud-optimized GeoTIFF (COG). Storing data in S3 cloud systems allows fast access and inspection of the data for users, providing an ARCO product.

## RESULTS

This section summarizes the most relevant results obtained from this work; however, implementation details and additional results are available as a Jupyter notebook at https://github.com/openlandmap/scikit-map/tree/feat_tsirf. The section mainly focuses in describing experiments and specification used to produce the results, leaving the discussion of them to the next section. Following the same order of contents of the methods section, we first report computational performance of TSIRF, time-series reconstruction accuracy of SWA and compared methods and, finally, the results of the Landsat bimonthly aggregated and reconstructed products.

A benchmark data set was designed to validate and compare time-series reconstruction methods. In particular, the land cover dataset ESA CCI (*European Space Agency, 2021*) was used in this scope. The initial 37 classes were aggregated into 18 classes based on similarity (water bodies classes were excluded from this analysis). This aggregation process was applied to maps from 2000 to 2020 to determine the number of changes in land cover within each pixel. Most of the pixels (approximately 95%) exhibited no changes in land cover during this period. To address this an additional strata was created for pixels with one or more land cover changes. Unchanged pixels were grouped into strata based on their corresponding aggregated ESA CCI class, resulting in a total of 18 strata for stable land cover and one strata for land-cover changes. We collected 2,600 random points from each stratum, ensuring a confidence level of 0.99 and a maximum standard error of 0.02. The selected points are shown in Fig. 4. From the Landsat bimonthly aggregated dataset described in the following section, bimonthly time-series were extracted for each sample point from 1997 to 2022, following stratification and random sampling.

### Comparison of discrete convolution computational backends

The three strategies to perform the discrete convolution presented in "Efficient Computation of Circular Convolutions" were compared in terms of computational time as

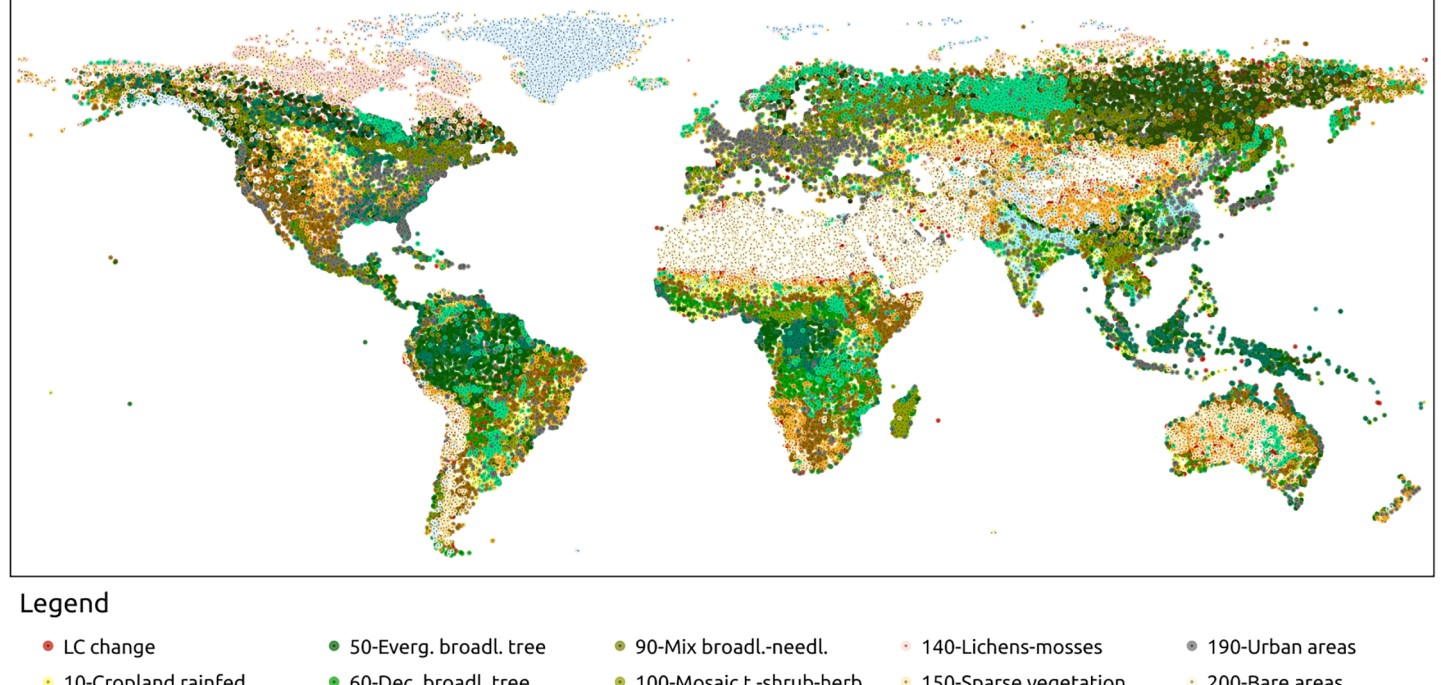

### Legend

- ● LC change
- ● 10-Cropland rainfed
- ● 20-Irrigated crop
- ● 30-Mosaic crop-nat.veg.

- ● 50-Everg. broadl. tree
- ● 60-Dec. broadl. tree
- ● 70-Everg. needl. tree
- ● 80-Dec. needl. tree

- ● 90-Mix broadl.-needl.
- ● 100-Mosaic t.-shrub-herb.
- ● 120-Shrubland
- ● 130-Grassland

- ● 140-Lichens-mosses
- ● 150-Sparse vegetation
- ● 160-Flooded tree cover
- ● 180-Flooded shrub-herb.

- ● 190-Urban areas
- ● 200-Bare areas
- ● 220-Permanent snow-ice

**Figure 4 Sampling points of the benchmark dataset.** The points were selected based on a stratified sampling design was based on aggregated version of ESA CCI land cover classes (*European Space Agency, 2021*). The initial 37 classes were aggregated into 18 classes. Pixels with stable LC over 2000 and 2020 are assigned to one stratum associated to an aggregated class. In addition, pixels with one or multiple LC changes were assigned to two additional strata, for a total of 19 strata. About 2,600 points per strata were selected for a total of 51,978 sampling points. The points where uniformly selected along the strata. However, point density could vary toward space due to presence of more localized strata.

a function of the three parameters that could influence it. In particular, the involved parameters are the time-series length $N_s$, the number of processed time-series $N_r$, corresponding to the number of rows in the matrix $V$, and the size of the convolution kernel. For the latter, considering the assumption $N_p < N_s > N_f$, implying $N_w < 2N_s + 1$, and that the ratio between $N_w$ and $N_s$ impacts the sparsity of the matrix $W_e$, we introduce the sparsity parameter $S$. This parameter is used to create a benchmark experiment in which random data are used to create $V$, $w_p$, $w_f$ and $w_0$, where the size of $w_p$ and $w_f$ is $\lceil SN_s \rceil$, with $S$ in the range $(0, 1)$ and $\lceil \rceil$ the ceil function. With $S = 1$, $W_e$ is a dense matrix. This experiment was run with each backend, *Matrix*, *Summation* and *FFT*, for each combination of $N_s = [1 - 2{,}200]$, $N_r = [1 - 40]$, and $S = [0.05, 0.1, 0.5, 1.0]$, repeating for three iterations each and averaging the total computational time for each combination of parameters and backend (for a total of about three million runs). The results were used to find the fastest backend at each point of the parameter space. The computation was performed on a workstation HP® Z840 with Intel® Xeon® CPU E5-2650 v4 @ 2.20 GHz × 48 and 128 GB of Hynix® DDR4 Synchronous Registered (Buffered) 2,666 MHz RAM.

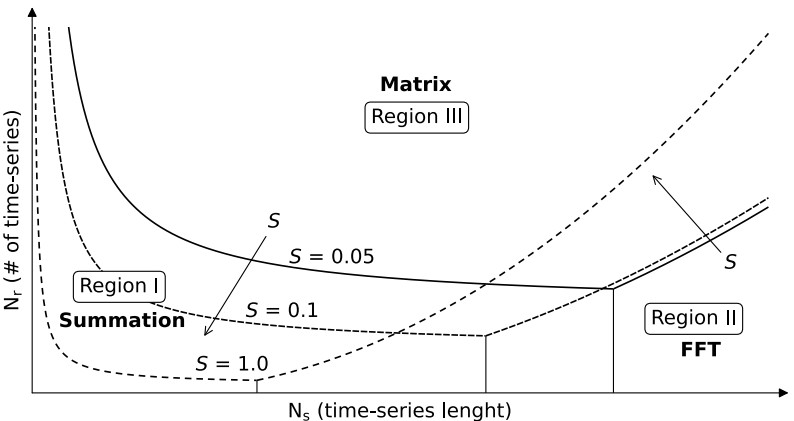

**Figure 5 Selection of the TSIRF backend with lowest computational time in function of the application.** The considered parameters are the number of samples in the time-series $N_s$, the number of time-series $N_r$ (*e.g.*, the number of pixels pixels) and the sparsity of the extended convolution kernels $S$ (fraction of non-zero elements). The curves were obtained by studying the asymptotic behaviors of computational complexity and by empirically fitting them to real data. Note that the figure does not report precise values for $N_s$ and $N_r$ since those greatly vary between different architectures and installed libraries, and the user is invited to perform a small test to orient to it. In general, when $N_r$ is very high, the Dense backend is probably the best choice, while it is the FFT backend in case of high $N_s$. When neither of the two is very high $S$ is small, the best option is the Sparse backend.

Since the results heavily depend on the used computational infrastructure and the used library, we combined the asymptotic behavior of each method and manual tuning of constant to fit functions that can describe the observed results. Those functions were used to produce Fig. 5 that helps users predict which backend works better depending on the application. The figure shows curves delimiting regions where each backend performs better, but without specifying the axis value, since this would be nonsensical for different computational infrastructures. However, since the asymptotic behavior should instead hold, shape and behavior of the regions should generally hold, and serve as baseline to decide which backend to use. The parameters $N_s$ and $N_r$ are on the *x*-axis and *y*-axis, respectively, and the three sets of curves are plot for $S$ equal to 0.05, 0.1 and 1.0. *Region I* is the region in which the summation backend is the fastest and shrinks toward the origin of the axis (low $N_s$ and low $N_r$) with increasing $S$. *Region II*, associated with the FFT backend, is located on the bottom right (high $N_s$ and low $N_r$) and expands with increasing $S$. Finally, for intermediate values of $N_s$ and high $N_r$ we generally fall in *Region III*, associated with the matrix backend. The selection of the best backend depending on the application is discussed in "Feasibility of Petabyte Scale Remote Sensing Time-Series Processing".

## Time-series reconstruction methods comparison

The benchmark dataset previously described was used to assess the effectiveness of time-series reconstruction methods under variable gap conditions by introducing artificial gaps into the extracted Landsat bimonthly time-series. These artificial gaps, also represented by "no data" values, were randomly inserted into the time-series to simulate

real-world data loss. By varying the number of artificial gaps, the performance of different reconstruction methods under increasing gap density was evaluated, to test the robustness and project the error to the original gap fraction. The evaluation was done by comparing the imputed values and the original values corresponding to the artificial gaps. This comparison enabled the determination of the ability of each method to recover the original land cover dynamics despite the presence of both cloud-induced and artificial gaps. Note that, to have a fair comparison, the optimization of the method's parameters was performed on a different dataset (*Consoli et al., 2023*).

We compared the performance of five different methods: (i) piecewise linear interpolation between available samples, in the following labeled *Piecewise LI*, (ii) TSIRF version of linear interpolation between available samples, as defined in Table 2, labeled *TSIRF LI*, (iii) TSIRF version of most recent images propagation smoothed with Savitzky–Golay filters, labeled *MR-SG*, (iv) SWA with the parameters used for production, labeled *SWA*, and (v) SG smoothed version of SWA, labeled as *SWA-SG*. The *Piecewise LI* was implemented imputing the value of the linear interpolation between the most recently available samples in the past and in the future of each missing value. For the other TSIRF-based methods the convolution kernel was set using the function described in the guidelines. For all the reconstruction methods, no weights for the future were used. However, for the SG a polynomial of order two was used to smooth a window of five samples centered on the third sample with zero-padding, making it non-causal (but only locally). An alternative could be to use non-symmetric SG by shifting the SG coefficients toward the $w_p$ vector. The SG filters were applied on top of SWA and MR results to evaluate the impact of smoothing.

We computed the Root Mean Square Error (RMSE) for each method in 10 experiments adding a varying fraction of artificial gaps. The results are reported for the Landsat NIR and red bands and for NDVI. For the NDVI, we also compared the impact of computing the NDVI first and then applying the reconstruction methods with the case in which the raw bands are first reconstructed and then used to compute the NDVI. The results are shown in Fig. 6. In the supplementary computational notebooks the same results are reported for all the bands. Projecting the curves toward the original gap fraction, for all the four cases, SWA and its smoothed version show a RMSE between 10% and 14% lower then the other methods. However, since the comparison only involves TSIRF-based or simple methods like piecewise linear interpolation, this comparison is not intended to comprehensive, but mainly used to compare alternatives for the Landsat aggregates reconstruction.

Furthermore, we compared the coefficient of determination ($R^2$) for different methods and different bands, similarly to what is done in *Hermosilla et al. (2015)*. Figure 7 shows the density scatter plot of the original NDVI against its reconstructed version using SWA on a the benchmark dataset with 10% of artificial gaps. The same plot is available for the other methods in the supplementary computational notebooks. In particular, the $R^2$ of each method in this experiment was 0.87 for Piecewise LI, 0.88 for TSIRF LI, 0.89 for MR-SG, 0.91 for SWA and 0.91 for SWA-SG. Notably, the ranking of the $R^2$ values is similar to the one of RMSE.

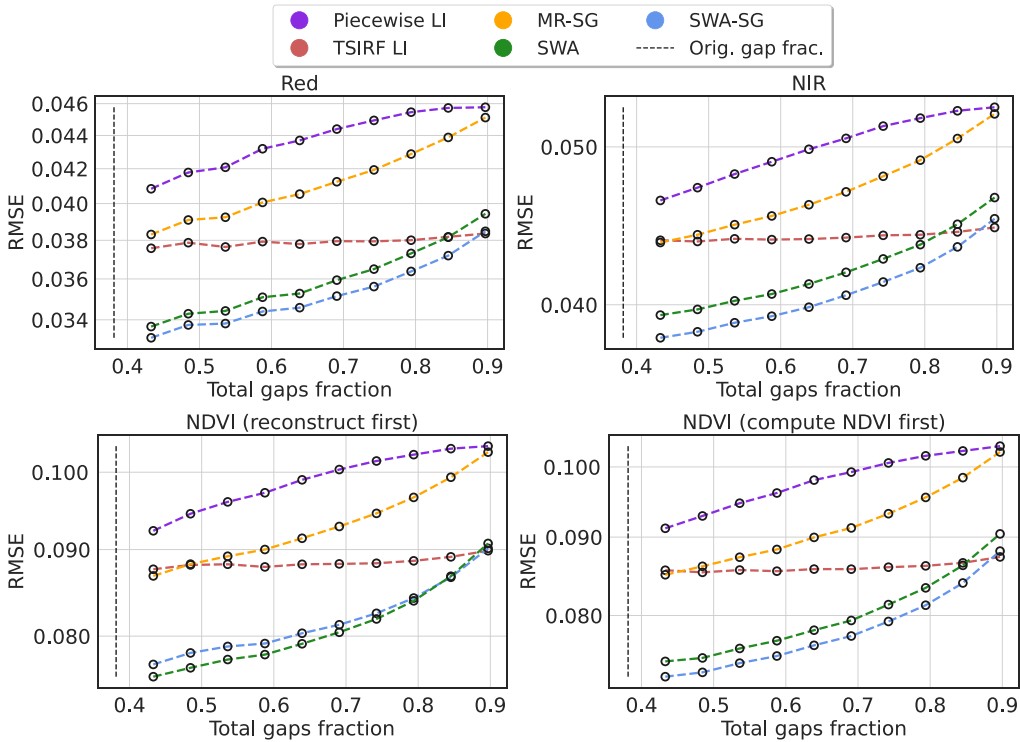

**Figure 6 Comparison of time-series reconstruction performance of different methods based on RMSE.** The error is computed on artificially created data gaps randomly located in the original time-series of Landsat bimonthly aggregated collection. Different fractions of artificial gaps are added to the original time-series in order to study the resilience of the method to gap presence and to project the RMSE to the original gap fraction. The compared methods are piecewise linear interpolation (Piecewise LI), its approximated version (TSIRF LI), approximated most recent image propagation with Savitzky–Golay filtering (MR-SG), seasonally weighted average (SWA), and SG filtered version (SWA-SG). The results are reported, from top to bottom, for near-infrared band and red band, NDVI computing the NDVI before performing the reconstruction and *vice-versa*.

## Land-cover and spectral dependency of time-series reconstruction error

To analyze the performance of the the time-series reconstruction for different LC, the RMSE was computed for each strata from the aggregated ESA CCI classes and the LC change samples for each band. In this case, only the results for a 10% of additional gaps are considered. Figure 8 shows the result for *SWA* with the previously described input parameters. The rows, representing the different LC classes, are ordered based on the total fraction of gap present in the input dataset, to see the correlation between RMSE and the number of gaps (except the LC changes class that is placed on the bottom of the graph). The error in each band is represented by a different symbol/color. The scatter plots and $R^2$ values are reported in Fig. 9. Finally, different error metrics in are computed for each band and summarized in Table 3, where also normalized versions of the RMSE and Concordance Correlation Coefficient (CCC) are included. The results were produced using the data available at https://zenodo.org/records/11150343 processed with the open-source

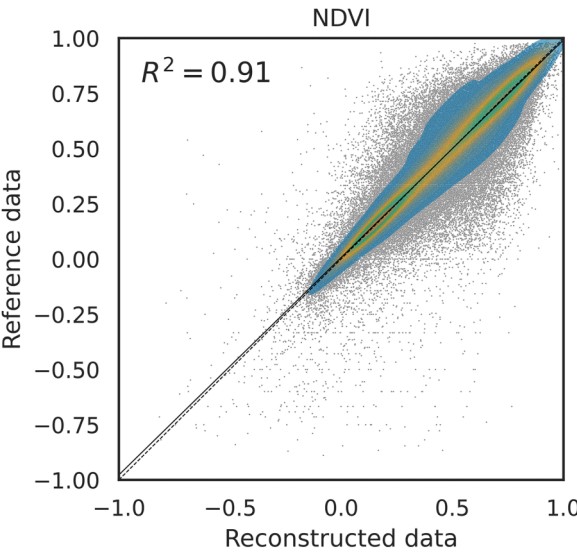

**Figure 7 Density scatter plot of NDVI data (reconstruct first) for 10% artificial gap fraction with SWA.** Similar patterns are observed for other methods, and not reported here, but variation are observable in the $R^2$ values. In particular we have 0.87 for Piecewise LI, 0.88 for TSIRF LI, 0.89 for MR-SG, 0.91 for SWA and 0.91 for SWA-SG.

code available at https://github.com/openlandmap/scikit-map/blob/feat_tsirf/. Note that the notebooks uses a pure Python implementation of TSIRF in order to facilitate reproducibility of the results. The production code based on hybrid Python/C++ code is also available on other branches of the same folders, but requires to be compiled before usage.

## Landsat bimonthly cloud-free global ARCO

This section reports some results of the GLAD Landsat ARD processing and reconstruction. Figure 10 shows a zoomed-in RGB composite of the passage from four cloud screened images to a bimonthly aggregated and reconstructed images of a tile in Papua New Guinea, summarizing the passages for the production of a single time-frame of one tile.

Figure 11, on top, shows a region of the global mosaic of the NDVI cloud-free ARCO product relative to the bimonthly timeframe January–February 2022. From the global mosaics, three zoomed-in regions, corresponding to one 16th of the original tiles, are analyzed to show details and artifacts. In particular, site *A* is located in Turkey (30E, 37N), site *B* in Papua New Guinea (144E, 7S) and site *C* in Norway (13E, 61N). For each site, in the bottom of the figure, we find from left to right the Landsat bimonthly aggregated NDVI (with no-data in gray), the Landsat bimonthly reconstructed NDVI, the QA layer associated to the reconstruction and the high-resolution RGB composites from Google Hybrid, used to compare with some reference data. In these last images, we also locate with a red cross the central location used as query point to analyze the associated time-series reported in Fig. 12. The values already available in the Landsat bimonthly aggregated NDVI are plotted with orange crosses in function of time from the year 1997 to the year

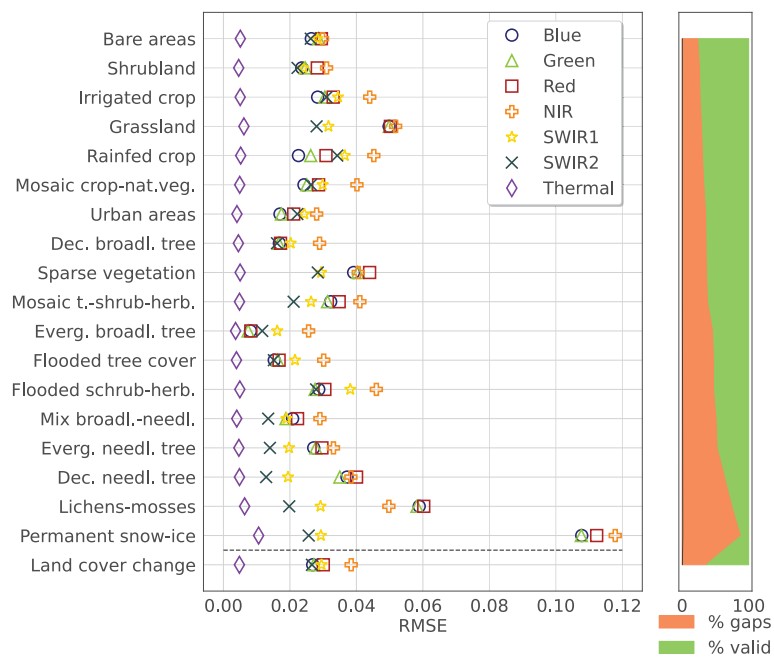

**Figure 8 Time-series reconstruction performance of SWA, the method presented in this work, for different strata (associated with ESA CCI land cover classes) and different Landsat spectral bands.** The error was computed adding a 10% of artificial data gaps to the Landsat bimonthly aggregated collection and calculating the RMSE on those samples. The different land cover classes are sorted based on the total fraction of data gaps (shown on the right side of the figure), while the error on pixel with varying land cover between 2000 and 2020 is shown in the last row. Each band is distinguished by a different marker and a different color as specified in the legend.

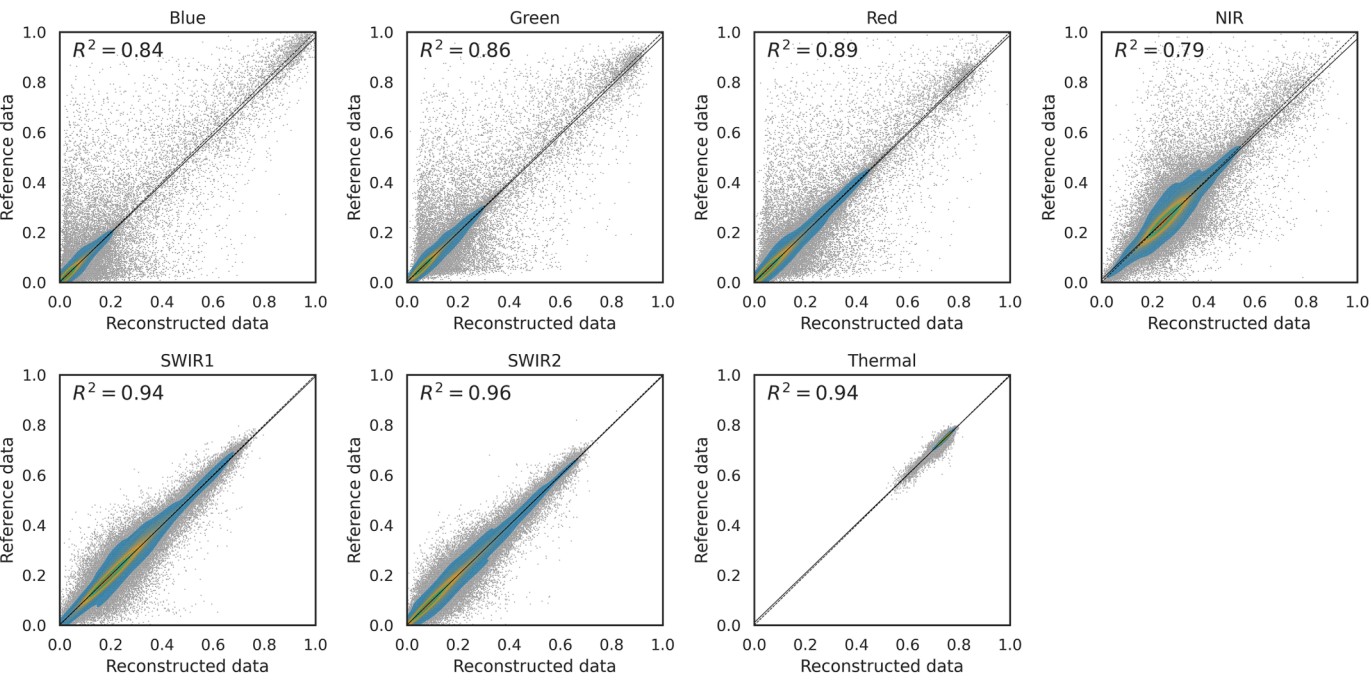

**Figure 9 Density scatter plot for different Landsat surface reflectance bands for 10% artificial gap fraction with SWA.** Different distributions and $R^2$ values are observable for different bands, in accordance to what is observed in Fig. 8.

**Table 3 Different reconstruction error metrics on each band for SWA with 10% of artificial gaps.**

| Band: | Blue | Green | Red | NIR | SWIR1 | SWIR2 | Thermal |
|---|---|---|---|---|---|---|---|
| RMSE | 0.031 | 0.031 | 0.034 | 0.039 | 0.028 | 0.024 | 0.005 |
| $RMSE/\mu$ | 0.56 | 0.33 | 0.32 | 0.15 | 0.12 | 0.15 | 0.0067 |
| $RMSE/\sigma$ | 0.39 | 0.37 | 0.33 | 0.46 | 0.25 | 0.2 | 0.25 |
| $R^2$ | 0.85 | 0.86 | 0.89 | 0.79 | 0.94 | 0.96 | 0.94 |
| CCC | 0.92 | 0.93 | 0.94 | 0.89 | 0.97 | 0.98 | 0.97 |

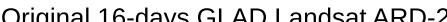

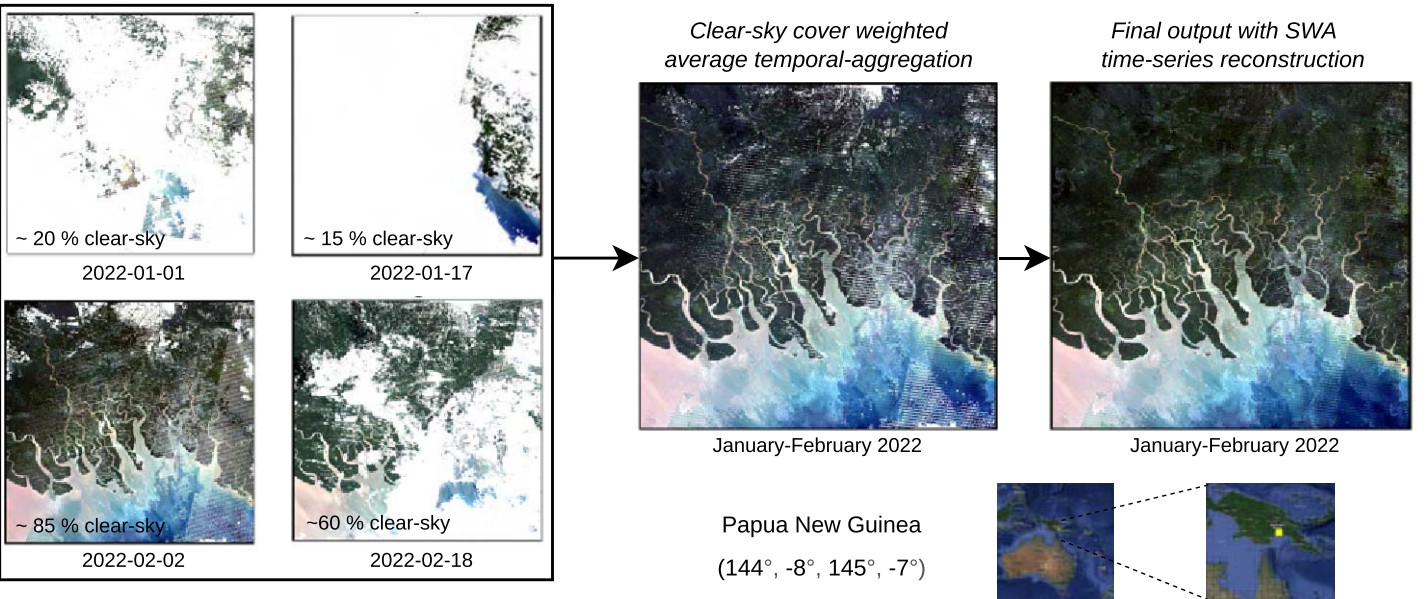

**Figure 10 Example with original 16-day Landsat scenes (percentages indicate cloud coverage) and final bimonthly cloud-free product for a small area in Papua New Guinea.** Notice some artifacts (lines) are still visible in the output.

2022, for a total of 156 samples per time-series. The Landsat bimonthly reconstructed NDVI time-series is instead plotted with a blue line. Due to the usage of only images from the past, it is possible to observe missing values in the reconstructed time-series in case of complete lack of images at the beginning of it. A green box highlights the time frame associated with the images shown in Fig. 11.

In Fig. 13 we compared the reconstructed Landsat images against Sentinel-2 L2A. We selected three locations in Europe with different latitudes and landforms. To each location we associated a different spectral band, in particular, the blue band with a location in Norway, the NIR band with a location in France and the SWIR2 band with a location in Italy (top-left and bottom-right corner coordinates are reported in the figure). The first location is $1° \times 1°$ in size, while the other two $0.5° \times 0.5°$, in order to show additional details and artifact. Aggregated Sentinel-2 images to the time-frames (Q1) 2019/12/02-2020/03/20, (Q2) 2020/02/21-2020/06/24 (Q3) 2020/06/21-2020/09/24 (Q4)

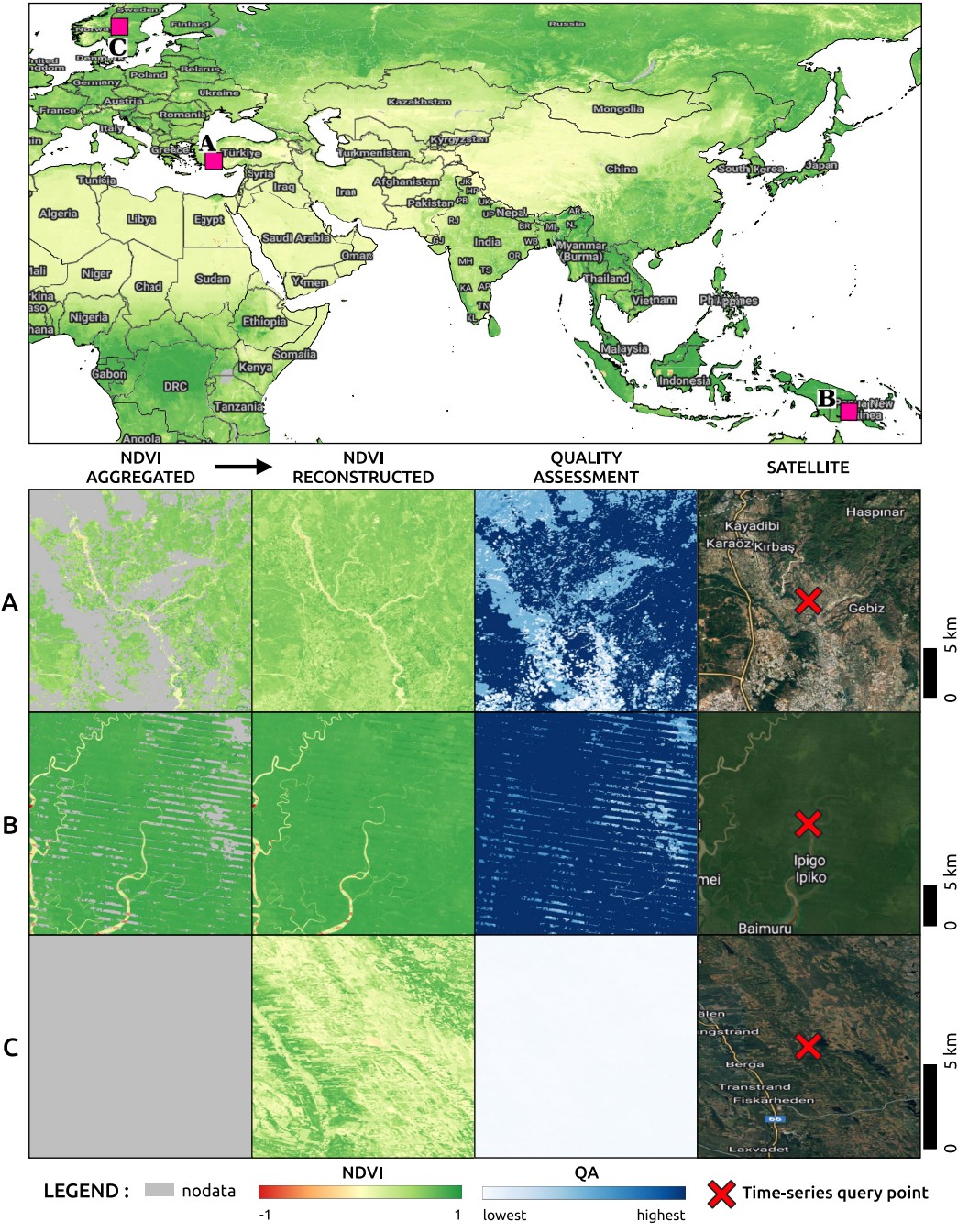

**Figure 11 On top the image shows part of a global mosaic of the Landsat bimonthly reconstructed and landmasked NDVI for the timeframe January–February 2022.** For three selected sites (A, B and C) zoomed in areas of 1001 by 1001 pixels are visualized in the bottom. For each of them, the first column shows the Landsat bimonthly aggregated image used as part of the input for the time-series reconstruction. The second column shows the reconstructed image and the third column the quality assessment relative to the reconstruction derived from the weights of the available images in the time-series used during for the reconstruction. The last column shows the satellite images derived from Google Hybrid with a cross in the query point of the pixels for which Fig. 12 report the whole time-series.

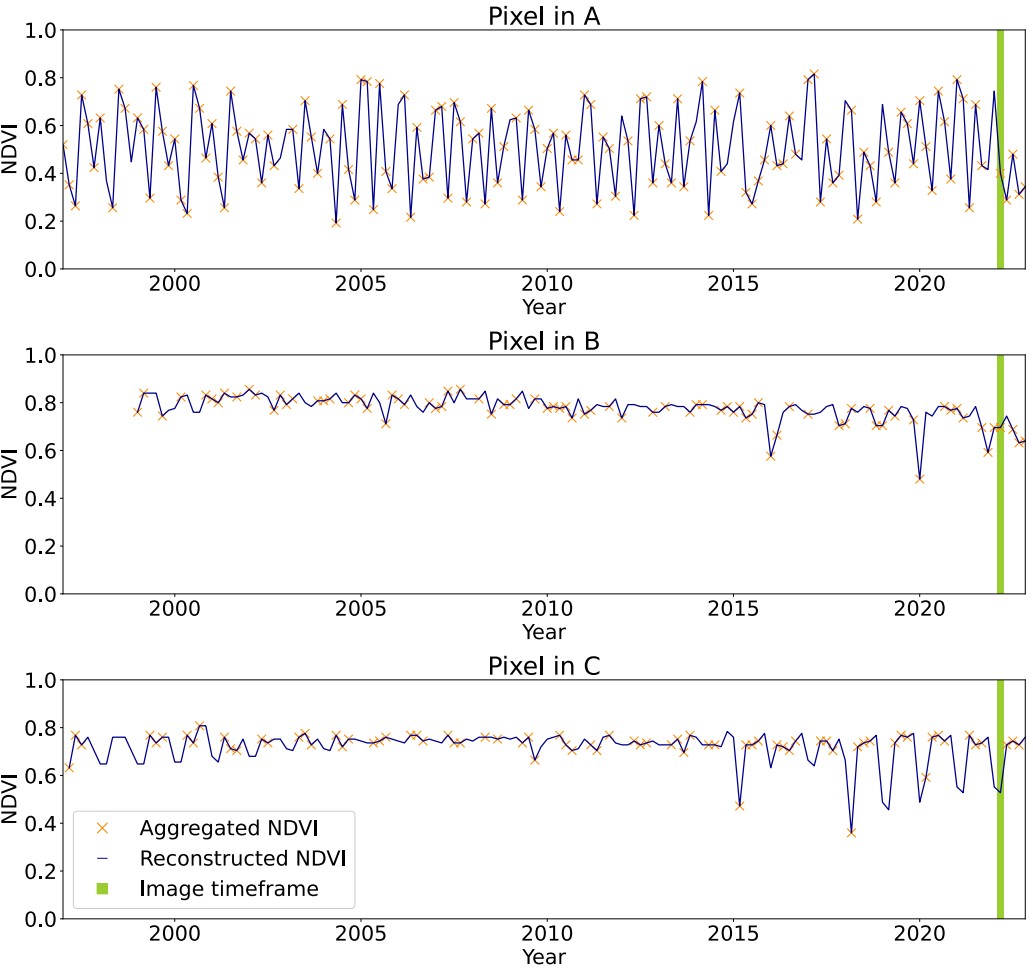

**Figure 12 Time-series of aggregated and reconstructed Landsat bimonthly NDVI for the pixel associated with the query points shown in Fig. 11.** In green the timeframe of Jan–Feb 2022, corresponding to the images in Fig. 11. The pixel in A shows NDVI seasonality and few missing values in the aggregated product. The pixel in B shows high and stable NDVI with missing values at the beginning of the time-series due to absence of satellite images before 1999 and the usage of causal time-series reconstruction method. Pixel in C shows seasonality only in the end of the time-series, probably due to systematic absence of images during winter season at the beginning of the time-series.

2020/09/12-2020/12/01. In order to have most overall with the Sentinel-2 time-frames, Landsat images are referred to the bimonthly time-frames Jan–Feb, May–June, Jul–Aug, Sep–Oct of 2020. To test reconstruction performance, for the corresponding time-frame of the visualization was considered as gap for all pixels. Since NIR and SWIR2 values are quite skewed non linear color-maps are used for visualization. It is evident that the quality of the blue band of Landsat is not comparable with the one of Sentinel-2 images, while NIR and SWIR2 bands mostly match. In the Landsat NIR image for the Jan–Feb time-frame it is possible to noticed some stripes artifacts propagated during time-series reconstruction. The Sentinel-2 images used for comparison are available at https://stac.ecodatacube.eu (*Witjes et al., 2023*).

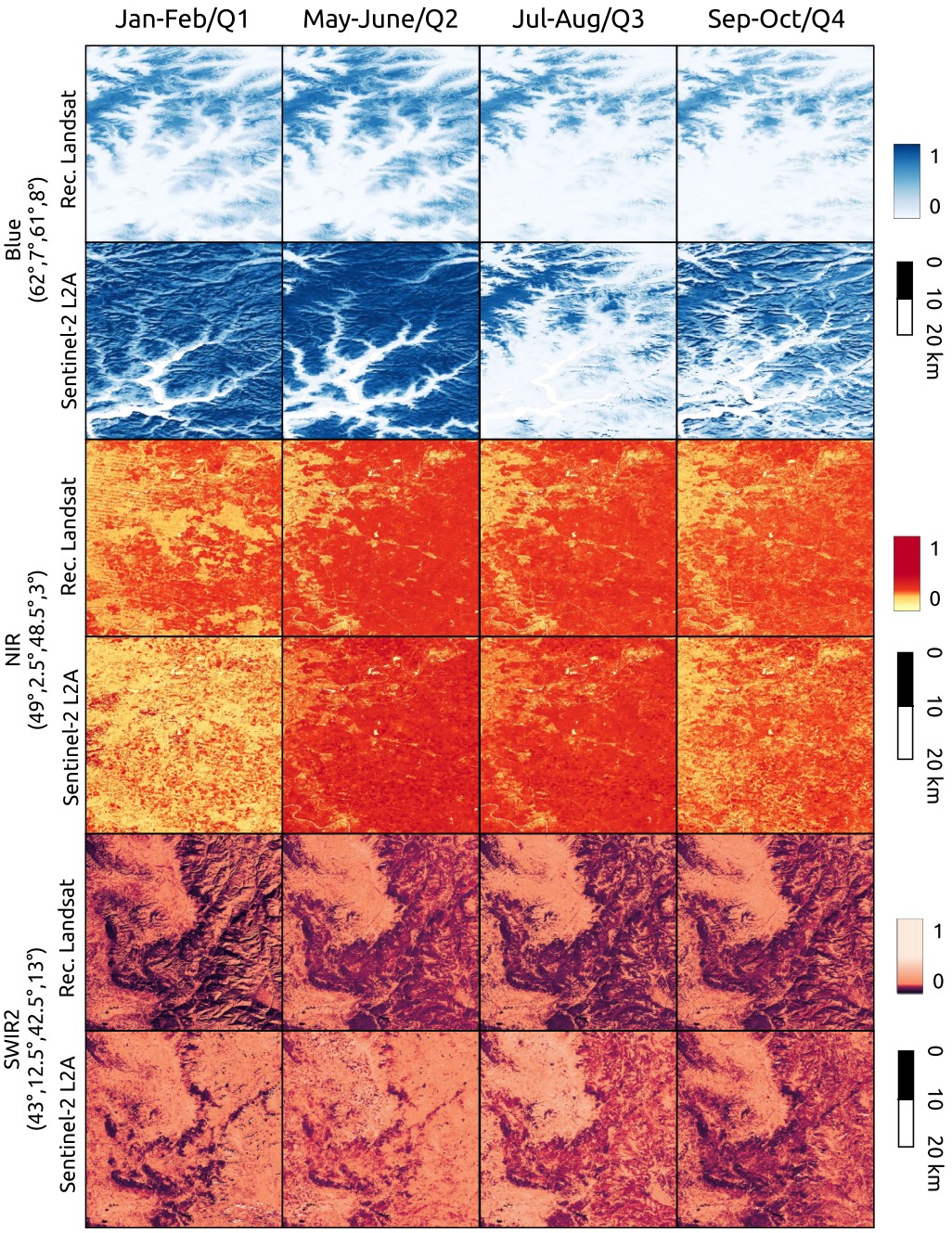

**Figure 13 Comparison between reconstructed Landsat and quarterly aggregated Sentinel-2 L2A images for different bands and locations.** Landsat images are referred to bimonthly time-frames of 2020. Aggregated Sentinel-2 images to the time-frames (Q1) 2019/12/02-2020/03/20, (Q2) 2020/02/21-2020/06/24 (Q3) 2020/06/21-2020/09/24 (Q4) 2020/09/12-2020/12/01. Each Landsat image was reconstructed with SWA after creating gaps in all pixels for the relative time-frame.

Finally, two complete global mosaics, corresponding to the January–February and July–August timeframes of 2022 for NDVI are available at https://github.com/openlandmap/scikit-map/blob/feat_tsirf/CATALOG.md. The values range from 0 to 250,

where 0 corresponds to −1 NDVI values and 250 to +1 NDVI value. In "Discussion" we discuss the cause of the artifacts present in the data and possible solution to attenuate them that will be developed in future works. That the land-mask is derived from the global land cover maps produced form the GLAD group in *Potapov et al. (2020)*, considering as *not-land* any pixel associated to a water-related class at least one time in the time-series from 2000 to 2020. Note that since the LC map is derived from the first version of the GLAD Landsat ARD the arctic regions and other smaller ones are not included in the masked and mosaiced data, but still available in the tiled products.

## DISCUSSION

### Feasibility of petabyte scale remote sensing time-series processing

The storage size of several collections of satellite images today reaches the petabyte scale. In addition to the Landsat collection mentioned above, as of August 2023, the Sentinel-2 archive amounted to a storage size of 20 PiB (*Bauer-Marschallinger & Falkner, 2023*; *Radeloff et al., 2024*). In addition, both missions keep generating about 0.5 PiB and 2 PiB of data per year, respectively (*Frantz, 2019*), without considering the distribution of different user-levels of the same product. Similar storage sizes are also required for satellite images used to monitor atmospheric conditions and solar activity, such as the ones produced by the Geostationary Operational Environmental Satellites (GOES) program. Indeed, as of July 2022, the GOES-16 and GOES-17 archives require more than 4.7 PB of storage size (*Willett et al., 2023*). While this does not represent a problem for most academic institution or big IT companies with high performance computing (HPC) infrastructure and programming know-how, the usage of such data is prohibitive for organizations with limited resources. Even more limited is the hosting of the data since it requires distributed storage systems and high-performance interconnect solutions. Platforms like Amazon Web Services (AWS), Microsoft Azure (Azure), and Google Cloud Platform (GCP) can help overcome hardware limitations. However, cutting-edge-derived products also require the usage of highly optimized and parallel software. Unfortunately, few open-source libraries provide a computational framework that can be used for such large-scale geospatial data applications. For these reasons, we developed scikit-map (https://github.com/openlandmap/scikit-map), a Python library that uses *ad-hoc* developed C++ functions for a computationally intensive task by directly exposing the data structures between the two languages, zero-copy operations and performing most of the operations with high parallelism. The library includes the described implementations of TSIRF, both with Python and C++ backends. In summary, *scikit-map* tires to combine computational performance with flexibility for large scale EO data processing, focusing in particular to time-series reconstruction.

In the result section, we compared the computational performance of the three algorithmic backends to perform the circulant convolution operation. In the following sections, we discuss the reasons and implications of this result, following the Bachmann–Landau notation for the asymptotic analysis. The computational complexity of the *matrix* backend is dominated by the matrix-matrix product(s). Most of the available libraries for matrix operations are based on BLAS (Basic Linear Algebra Subprograms) that

chose the best algorithm based on the computational architecture and sizes of the involved matrices for each operation. However, for most cases $V$ is a rectangular matrix for which fast methods for the matrix-matrix product are convenient only for very large-scale and specific cases (*Knight, 1995*), the computational complexity of the selected algorithms is likely asymptotically bounded by $\Theta(N_r N_s^2)$. For the *FFT* backend instead, we know that for $N_s \to \infty$ the complexity of the convolution of each time-series scales asymptotically as $\mathcal{O}(N_s \log_2 N_s)$ (and the same goes for $\lceil S \rceil$) (*Flannery et al., 1992*), while for $N_r \to \infty$ as $\mathcal{O}(N_r)$. Finally, the *summation* backend is probably asymptotically bounded by $\Theta(N_r \lceil S N_s \rceil^2)$. This analysis helps to understand the results presented in Fig. 5. For example, for $N_s \to \infty$, the *FFT* backend scales as $\mathcal{O}(N_s \log_2 N_s)$, while the *matrix* and *summation* ones as $\mathcal{O}(N_s^2)$, making the first the best choice for processing of time-series with many samples in time (*Region II* in the figure), as in the case of historical data from GOES. On the other hand, for $N_r \to \infty$, all methods behave asymptotically as $\mathcal{O}(N_r)$. However, from empirical results, the *matrix* backend is generally the best option (*Region III* in the figure). In this case, the dominant factor comes not from computational complexity, but from the fact that the BLAS code is highly optimized, and probably caching optimization allows it to reuse the content of $W$ for all time-series without reloading it from memory. As already mentioned, a scenario that falls into *Region III* is when the same processing is applied to several time-series, as for the case of a moderate and high spatial resolution EO dataset (MODIS, Copernicus Sentinel, and Landsat collections). Finally, for $S \to 0$, we clearly expect the *summation* backend to perform better, since its complexity reduces quadratically with $S$, while the computational complexity of the *FFT* backend only linearly and that of the *matrix* backend is not impacted at all. This defines the boundary of region *Region I*, which could be of interest when using Savitzky–Golay or other digital filters with small window length compared to time-series length.

The Python library `SciPy` also offer different backends to perform the convolution (https://docs.scipy.org/doc/scipy/reference/generated/scipy.signal.convolve.html). In particular, its backend *full* matches the backend *summation* in TSIRF, while the *FFT* one intuitively matches the *FFT* in TSIRF. However, there is no equivalent for the *matrix* backend, which was instead the most convenient option for our processing of the GLAD Landsat ARD archive. In addition, with the used software setup and computing infrastructure, even if using the same number of CPUs, we noticed that the computational efficiency of the `SciPy` library was not comparable to the one achieved by combining the `C++` libraries Eigen (https://eigen.tuxfamily.org/) for algebraic operations and OpenMP (https://www.openmp.org/) for multi-threading. This is probably due to limitations imposed by the Python's Global Interpreter Lock (GIL) and different implementation strategies.

## Time-series reconstruction for EO data imputation

Even if in literature several works used purely spatial approaches for EO data imputation (*Desai & Ganatra, 2012*), in this work, we only focused on time-series reconstruction methods (including hybrid approaches). This is because purely spatial methods risk

creating patches in the imputed images, reducing the actual spatial resolution of the final product. *Yu et al. (2021)* categorizes time-series reconstruction methods into three categories: local window, global window, and others. Differently, *Siabi, Sanaeinejad & Ghahraman (2020)* and others (*e.g.*, *Desai & Ganatra (2012)*, *Gerber et al. (2018)*, *Zeng, Shen & Zhang (2013)*) divided the methods between spatial, temporal, and spatio-temporal. Following this categorizations, our framework would fall under the categories *temporal* and *global window*, since it only uses the temporal information but consider the whole time-series to reconstruct each missing sample.

The mentioned articles, also provides a comprehensive review of imputation methods for EO data. However, only few of them represents a good option for reconstruction of petabyte size historical EO time-series, and most of them are not openly available. Similarly to what is done in *Julien & Sobrino (2018)*, to assent the performance of SWA, we decided to create a benchmark dataset based on the Landsat product that we presented in this work, since different time-series reconstruction methods could perform quite differently depending on the data set to which they are applied. Indeed, an intrinsic limitation of SWA, is that applying it to time-series where there is no seasonality (or known periodicity in general) would be actually equivalent to the numerical approximation of most recent available value propagation. As already mentioned, the benchmark dataset is openly available and we invite other groups to use it to compare the result with other methods.

From Fig. 6, projecting the error towards the original gaps fraction, we can observe that for all bands/indices, SWA, or its smoothed version, are the methods with best performance. In addition, we also wanted to study whether computing the NDVI before performing the reconstruction or doing the *vice versa* would have an impact on the final error. However, comparing the two graphs in the bottom, we can observe only a slight increase in error for the smoothed version of SWA. We are aware that this comparison is not comprehensive of space-time or data-fusion approaches that could potentially outperform SWA, however, since we did not find openly available code to compare such methods and we currently did not develop other methods that can be applied to large-scale dataset, we limited the analysis to what is reported. More comprehensive comparisons are available in *Kandasamy et al. (2013)* and *Julien & Sobrino (2010)*.

Figure 8 instead analyses the reconstruction-error on different LC classes for each of the 8 Landsat spectral bands. In this case, only the result relative to SWA with a 10% of additional artificial gaps is considered. The aggregated ESA CCI classes of the benchmark dataset are listed on the left. To help taking into account that different land covers could present a larger fraction of missing values, we used this parameter to sort the classes, in addition to plot it on the right of the figure. Observing the figure, it can be noticed that for most of the LCs, there is an inverse correlation between the RMSE of each band end their wavelength. This is probably due to the fact that bands with shorter wavelengths are more sensitive to atmospheric contamination, introducing more noise in the time-series. Indeed, considering the spectral absorption profile of snow/ice shows, it is evident that more blue light is reflected compared to other LC classes, reducing the relative noise in the satellite

sensor reading. It is also interesting to note that the error in samples with changes in land cover is not particularly high. This is probably due to the fact that because of the deforestation of the last years, many LC changes involve the transformation of forest pixels into crops or grassland pixels, for which the error is generally in the same order of magnitude. This also validates the efficacy of using a envelop attenuation for the weights of SWA to track LC changes.

## Analysis-readiness of EO data and Landsat processing

The analysis-readiness of the EO data is a trending topic of debate (*Baumann, 2024*; *Truckenbrodt et al., 2019*). The definition of the term is *per se* controversial. In fact, anyone could argue: "*Ready for what?*". Following the definition of the USGS (https://www.usgs. gov/landsat-missions/landsat-us-analysis-ready-data), their Landsat ARD products drastically reduce the magnitude of data processing required for direct use in monitoring and evaluating landscape change (*Dwyer et al., 2018*). The product includes a Quality Assessment (QA) layer that allows, for instance, to filter out cloudy pixels from the analysis. *Frantz (2019)* presents the FORCE (Framework for Operational Radiometric Correction for Environmental monitoring), designed to produce ARD Landsat and Sentinel data. The work highlights that the USGS Landsat ARD still amounts to prohibitive size data volume, in addition to suffering from poor spatial and temporal consistency, and attempts to address these issues. However, FORCE is only provided as a tool, and not as a global and historical consistent dataset. For this reason, in *Potapov et al. (2020)*, the GLAD group presents the already mentioned GLAD Landsat ARD, which represents a unique product in terms of consistency and usability of the data, to the best of our knowledge.

Evidently, the cloud presence could impact for the production of biophysical indices maps such as normalized differences, albedo, gross and net primary productivity (GPP and NPP), or water vapor. In addition, for applications like LU/LC classification, most machine learning modeling techniques (*e.g.*, random forests, artificial neural networks, support vector machines) could produce artifacts in the output map if the contaminated pixels are provided in input. An option is to label such pixels as no-data and eventually propagate the no-data value to the output. However, considering that, at the global level from 1997 to 2022, in the bimonthly aggregated Landsat product presented in this work, about 40.1% of the pixels are no data, propagating this to the output would mean to do not classify large areas. Yearly statistics of valid pixels fraction for each year is reported in Table 4, where is interesting to observe the increasing trend of images availability. A per-pixel count of the presence of data gaps in the Landsat bimonthly aggregated time-series is shown in Fig. 14, for which source COG data at 30 m resolution can be downloaded or visualized at https:// github.com/openlandmap/scikit-map/blob/feat_tsirf/CATALOG.md. In addition to higher gap presence in cloudy regions, it is possible to observe patterns determined by scenes overlap. However, all these statistics also depends on the aggregation factor and on the used flags criteria to define a pixel valid or not. For instance, to consider snow a valid pixel or to aggregate the product quarterly would reduce the amount of gaps. In any case, the percentage of gaps are not negligible, especially in some regions. In fact, to produce some products derived from Landsat data, the GLAD group applies time-series
**Table 4 Percentage of valid pixels for all tiles and all time-frames of each year in the Landsat aggregated product.**

| Year | 1997 | 1998 | 1999 | 2000 | 2001 | 2002 | 2003 | 2004 | 2005 |
|------|------|------|------|------|------|------|------|------|------|
| Valid pixels | 42.3% | 45.3% | 50.7% | 58.0% | 58.7% | 56.9% | 51.1% | 57.1% | 56.8% |
| | 2006 | 2007 | 2008 | 2009 | 2010 | 2011 | 2012 | 2013 | 2014 |
| | 58.1% | 58.1% | 57.5% | 60.3% | 57.2% | 56.5% | 52.7% | 63.7% | 67.1% |
| | 2015 | 2016 | 2017 | 2018 | 2019 | 2020 | 2021 | 2022 | **Mean** |
| | 67.8% | 68.7% | 67.5% | 67.5% | 67.4% | 68.3% | 68.0% | 67.8% | 59.6% |

reconstruction techniques such as piecewise linear regression (*Potapov et al., 2021*). This approach still requires to load and preprocess the time-series to use these images, again limiting the usage of the data for some user, in addition to potentially achieving poorer results in the time-series reconstruction when compared to other methods. Figure 15 compare reconstructed Landsat summer and winter images with averaged annual Sentinel-2B images and shows that our product still suffer from artifact presence, in particular in northern area or rain forests. In addition, it is possible to notice the Landsat 7 strips and the shapes of different scenes from which each image is produced (Fig. 11). Similar artifacts can be observed in Fig. 13. In future versions more efforts will be invested in reducing these limitations. Note that all the produced mosaics were landmasked using pixels classified as land-related classes, for at least one time frame, in the land-cover product described in *Potapov et al. (2022)*. In the boundaries between water bodies and land, depending also on the timeframe, some not masked pixels could actually be derived from water reflectance and show heavy contrasts compared to land pixels.

An important first achievement in processing is the reduction in the storage space of the derived products. In particular, compared to the 1.4 PiB of the input collection, aggregated and reconstructed bimonthly only require 60 and 100 TB of storage, respectively. Derived products of global time-series with the same temporal and spatial resolution saved as COG require approximately 20 TB each. However, to achieve such compression, the temporal interval increased from 16-days to bimonthly, for some applications such detection of multiple corp cycle during one year, this could be a limiting factor. Nevertheless, considering that globally for several regions in some part of each year there are anyway no available images, this does not necessarily impact the analysis at global scale. Another factor impacting the compression is the storage of data as 8-bit unsigned integers instead of 16-bit ones. This could be relevant, for example, for the mapping of different vegetation species distributions or soil moisture. With these considerations we would expect a total compression factor of about 7.7 times, while we observe a compression factor of about 14 times. Part of the compression comes indeed from the storage of the images as 1,024 by 1,024 compressed chunks instead of 1 by 4,004 chunks. This emphasizes the importance of storing and compression strategies. In addition, we released an yearly aggregated version using the percentile 50 for each pixel, also available at https://stac.openlandmap.org. For applications that do not require a temporal resolution superior to one year, this represents the most compact way to access the data, in addition to being less affected by artifacts.
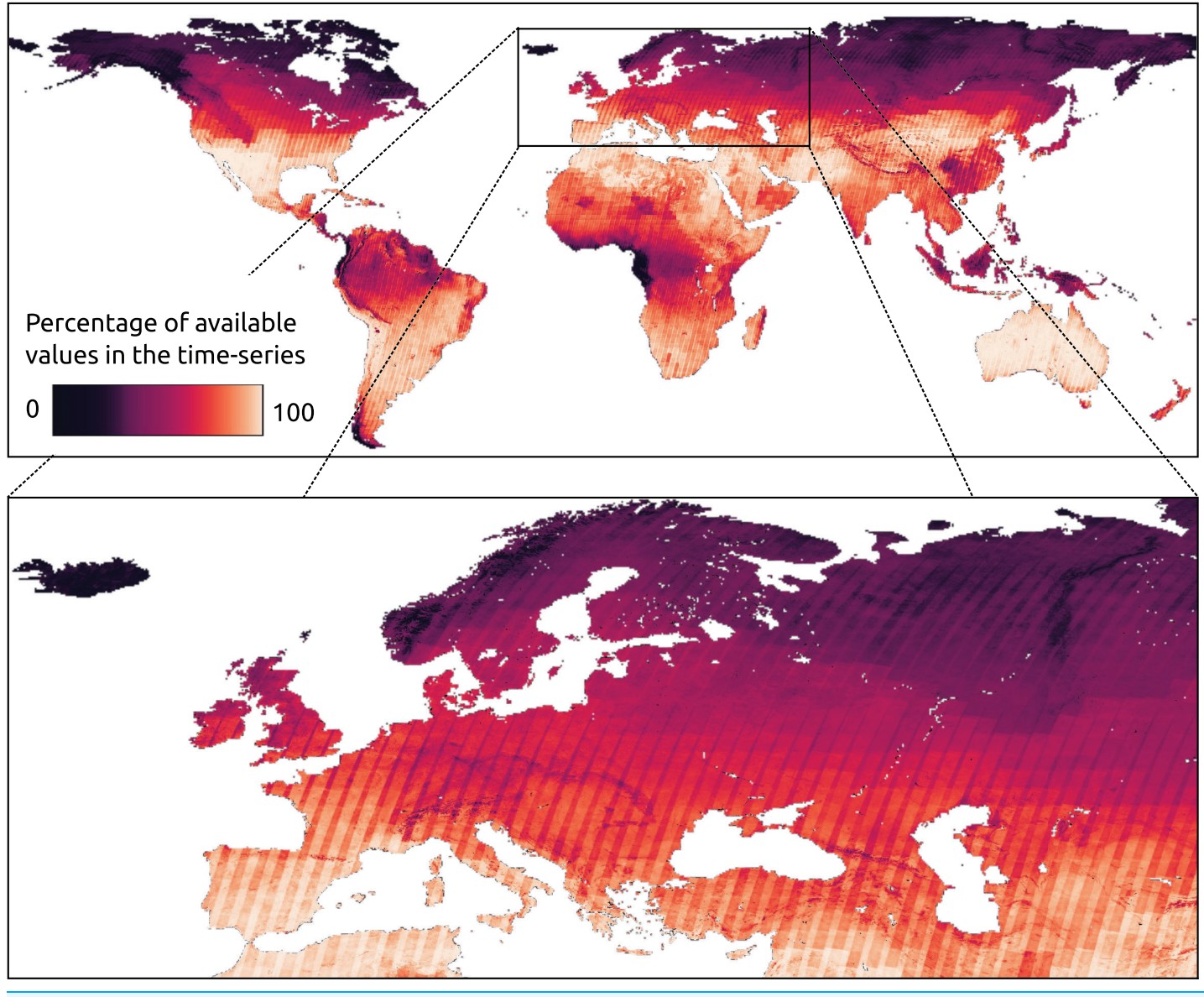

**Figure 14 Per-pixel count of available value in the bimonthly aggregated Landsat time-series from 1997 to 2022.** Since all the time series are of the same length and determined by the considered period and aggregation factor, we report this information as percentage of available values (not gap) in the time series. Darker areas are more affected by presence of data-gaps in the time-series. In addition to cloud-presence and snow-cover, it is possible to notice patterns determined by overlapping scenes in the original Landsat raw images.

Another problem connected to the time-series reconstruction strategy can be observed in the time-series of the point queried at site C shown at the bottom of Fig. 12. In this case, it can be noticed that the complete absence of available images during winter months in northern regions brings to the propagation of values from summer into them. This result in the shown case in an NDVI time-series for those regions that resembles the one of a tropical area as the one shown for cite B. To overcome this issue, it could be possible to do not consider as gap pixels flagged as snow and ice. Finally, as anticipated, the application of SWA without using values from the past implies that if no values are available at the

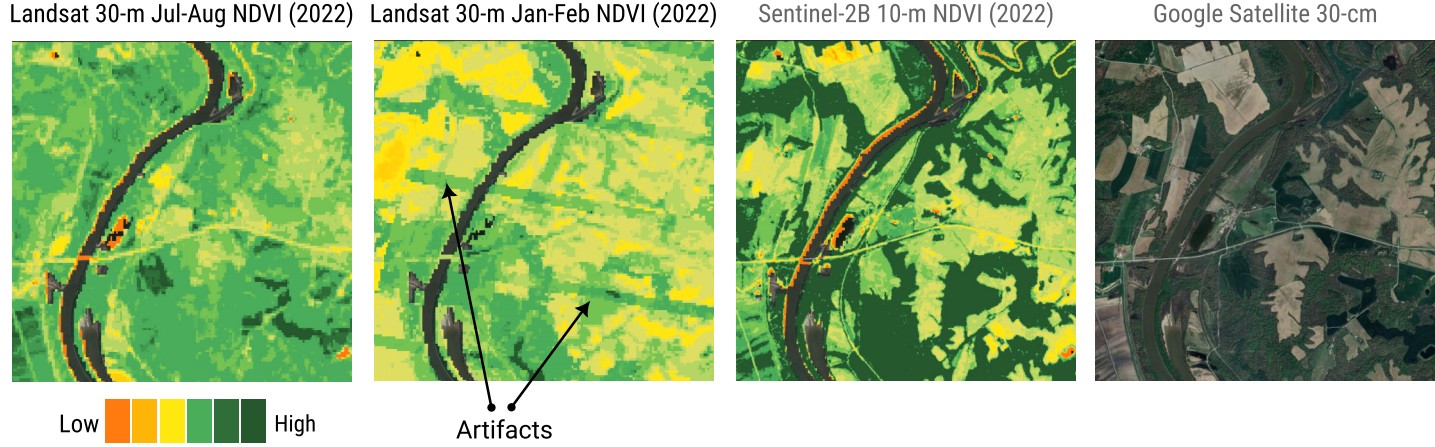

**Figure 15 Landsat *vs* Copernicus-2: comparison of Landsat bimonthly ARD NDVI at 30 m (this project) *vs* Sentinel 10 m spatial resolution annual NDVI for a smaller 5 × 5 km area near Ellis Grove in Illinois.**

beginning of the time-series, missing values will still be present. This could be solved, for example, by using future images only for such samples and assigning them to $w_f$ lower values compared to $w_p$. All these consideration will be taken into account for the production of the next version.

## Additional applications and future works

As for TSIRF, the Savitzky–Golay (SG) filters are a flexible tool that can be used for different applications by simply adjusting the input parameters. Through local polynomial fitting, SG filters smooth out noise while keeping the analytical signal features, thus, is widely applied for data smoothing and time-series reconstruction (*Chen et al., 2004*; *Savitzky & Golay, 1964*). By accurately calculating the elements of the convolution kernel of TSIRF, it is possible to combine several time-series reconstruction techniques with SG filters without the need to apply them in a second step, saving additional computational resources. As future development we plan to automatize the convolution kernel definition to integrate SG filter on top of other convolution kernels for arbitrary parameters.

Another approach to perform time-series reconstruction with TSIRF could be to set all the values of the convolution kernel using pure data-driven approaches. For instance, it could be possible to tune the convolution kernel using the gradient descent algorithm to minimize the RMSE on artificially generated data-gaps. Other loss functions or minimization algorithms could also apply. However, such an approach would lead to less interpretability about the application of the time-series reconstruction.

The flexibility of the framework, can also allow to extend it from purely temporal to a spatio-temporal approach (*Liu et al., 2019*). For instance, for each pixel time-series, it could be possible to create another time-series containing the value of the spatially nearest available pixel for each timeframe. These time-series can be concatenated to the extended ones in Eq. (4) (while maintaining the proper padding distance), and the convolution kernel defined to also include those values, possibly with lower weights for them compared to the original pixel time-series. A similar approach could also be used to perform data

fusion. Indeed, concatenating to the original one a time-series from an harmonized product and then performing the convolution could allow the use of information from other sensors while keeping the same spatial resolution.

An option to improve quality of the Landsat ARD bimonthly product could indeed be the data fusion with the MODIS collection, which has lower spatial resolution but shorter revisit period (8-day). However, this approach requires the usage of time-series harmonized in space, time and variable spectrum. Then, a modified convolution kernel based on SWA can be used, in combination with the concatenated and padded time-series. In particular, the part of the convolution kernel associated with MODIS data can be defined with same seasonal end envelope attenuations as the Landsat one, but scaled by a factor $f < 1$, such that higher priority will be given to Landsat values with the same seasonality and temporal distance from the value under reconstruction. This could help for instance to reduce the reconstruction artifacts in winter months in Northern regions. Since the result would still be a weighted average along the time-series that still involves 30 m resolution values from the Landsat data, by properly tuning the value of $f$, we expect that the presence of artifacts due to double resolution will be marginal. A combination of Landsat and MODIS satellite collections for data fusion was proposed in *Luo, Guan & Peng (2018)*. However, their product is unfortunately not openly available, so it is also difficult to validate and reproduce.

Another aspect that will be explored in future version is the usage of different projection strategies to further reduce the storage size. As demonstrated in *Bauer-Marschallinger & Falkner (2023)*, the projection used to store Sentinel-2 images increases about 33% the amount of information that they actually provide. *Bauer-Marschallinger, Sabel & Wagner (2014)* proposed a solution to the oversampling issues consisting of a girding system of seven different continental projections in order to minimize area deformation, from which the name "*Equi7 Grid*", reducing the global mean oversampling to about 2%. In addition, since the area deformation is minimal, data store in Equi7 can be directly used as input for application where the a match between graph-distance and geodesic-distance is required (*e.g.*, convolutional neural networks or hydrological modeling). However, this strategy has some drawbacks when working on a global scale. Since different projection systems are used, to visualize global mosaics requires different re-projection on each continent, with consequent computational over-head. In addition, applying spatial convolutions could be tedious in transitional regions, and padding strategies could be required. An alternative is to work with "Discrete Global Grid System" (DGGS) (*Kmoch et al., 2022*), where the spatial distribution of the data is obtained by associating each location with an equal-area cell from Earth surface tessellation. Nevertheless, this approach requires the usage of proper data-structure to store and process the data. In summary, both Equi7 and DGGS are elegant solutions for storing global EO data, but both require infrastructure and additional efforts to process the data.

Recent works targeting production of analysis ready geospatial images focus on usage multi-sensor data. This is done using different approaches, such as deep-features extraction (*Zhou et al., 2023*), multi-spectral data-fusion (*Moreno-Martínez et al., 2020*) and the creation of geospatial foundation models (*Jakubik et al., 2023*; *Han et al., 2024*).

Deep-learning (DL) with convolutional neural networks demonstrated effectiveness in extracting features from raw surface reflectance data suitable for LU/LC classification (*Dou et al., 2024*). Applying unsupervised or transfer learning techniques on such models, it is possible to extract abstract features from a set of EO images that can vary in acquisition time, spectral band or even satellite sensor and use them for different modeling tasks. The advantages of such approaches consist in the possibility to combine several raw EO data sources and compress their useful information in few general purpose abstract features. On the other side, the usage abstract features lead to lack of interpretability of the features importance ranking of a model. In addition, the abstract features can not be used to directly compute well established biophysical indices (*Montero et al., 2023*). This implies the usage of modeling techniques to produce such indices, while using the reconstructed bands produced in this work they can be computed from simple arithmetic formulas. Overall, deep-features extraction could be a promising enabling approach for several applications in EO data classification. Similarly, foundation models such the one proposed in *Jakubik et al. (2023)* are innovative and flexible options for several geospatial applications, including ARD generation. In future works could be interesting to compare complete satellite images produced by foundation models with the ones obtained with classical imputation methods. Another front that could be explored is the training of a foundation model based on a set of reconstructed images selected by expert based visual interpretation, taking advantage of both approaches and potentially avoiding some model biases. As a final note, we want to highlight that the boundary in the definition of ML (or DL) and other data driven approaches can be blurred. Indeed, if as previously proposed the weights of TSIRF are all obtained by minimization of a loss function, TSIRF could be also defined as ML model.

## CONCLUSION

Processing of petabyte size collections of EO time-series is a challenging task that requires the use of cutting-edge technologies. The ARD versions of such collections can rarely be used as direct input for widespread modeling techniques. The additional processing needed can be prohibitive in terms of computational cost, or lead to poor quality results when naïve techniques are used. Few options suitable for such a task are openly available and none are flexible enough to be easily tuned for different applications. With the TSIRF computational framework, we tried to fill this research gap. By properly setting its convolution kernel, different processing techniques can be obtained. Implemented with optimized and parallel code in `C++`, TSIRF can be applied to very large-scale time-series processing.

In addition to the TSIRF framework, this article also introduced SWA, a time-series reconstruction strategy integrated in it. The method takes advantage of the seasonality present in EO time-series to reconstruct missing values, while also giving higher priority to more recent images compared to the one under reconstruction. To assess the reconstruction performance of the method, a benchmark dataset was created adding

artificial missing values to the time-series, and computing the RMSE on those samples. The RMSE of SWA and other competing methods was computed for different surface reflectance bands and NDVI. For all cases, the best performing method was SWA or its version smoothed with Savizky–Golay filters, with at least a 10% error reduction compared to the other methods.

As first real-case application, TSIRF was used to process the entire GLAD ARD Landsat archive to derive a cloud-free bimonthly aggregated product. The temporal aggregation was performed by averaging four 16-days interval input images, weighting them by the local clear-sky factor. The remaining missing values in the time-series were then reconstructed applying SWA. Both the aggregation and the reconstruction were implemented within TSIRF. The reconstruction phase of the seven Landsat bands at global scale form 1997 to 2022, for a total of more than two trillions of 156 long time-series, requires about 28 hours of computation using 1248 Intel® Xeon® Gold 6248R CPUs. The reconstructed images can be used as input for ML models to derive biophysical indices, such as NDVI, mosaiced and stored as COGs, leading to an ARCO dataset. As a limitation, the global mosaics and the reconstructed time-series show some artifacts and evident outlier values. However, the product will serve as a baseline for the development of further improved versions. Hosting it as open data in cloud storage will require about 20 TB per band/index for the entire 30 m resolution, bimonthly interval, historical time-series, enabling its utilization to many EO data users.

## ACKNOWLEDGEMENTS

The authors thank Peter Potapov, Co-Director Global Land Analysis & Discovery group (GLAD), and the whole GLAD group for providing assistance with the Landsat ARD 16-day composites.

### Funding

This research was supported by a grant to the Land & Carbon Lab from the Bezos Earth Fund. The Open-Earth-Monitor Cyberinfrastructure project has received funding from the European Union's Horizon Europe research and innovation programme under grant agreement No. 101059548. The funders had no role in study design, data collection and analysis, decision to publish, or preparation of the manuscript.

### Grant Disclosures

The following grant information was disclosed by the authors:
Bezos Earth Fund.
European Union's Horizon Europe Research and Innovation Programme: 101059548.

## Competing Interests

Davide Consoli, Leandro Parente, Rolf Simoes, Murat Sahin, Xuemeng Tian, Martijn Witjes and Tomislav Hengl are employed by OpenGeoHub. Lindsey Sloat is employed by World Resources Institute (WRI).

## Author Contributions

- Davide Consoli conceived and designed the experiments, performed the experiments, analyzed the data, prepared figures and/or tables, authored or reviewed drafts of the article, and approved the final draft.
- Leandro Parente conceived and designed the experiments, analyzed the data, authored or reviewed drafts of the article, and approved the final draft.
- Rolf Simoes conceived and designed the experiments, performed the experiments, analyzed the data, prepared figures and/or tables, authored or reviewed drafts of the article, and approved the final draft.
- Murat Şahin analyzed the data, prepared figures and/or tables, and approved the final draft.
- Xuemeng Tian performed the experiments, analyzed the data, authored or reviewed drafts of the article, and approved the final draft.
- Martijn Witjes conceived and designed the experiments, analyzed the data, prepared figures and/or tables, and approved the final draft.
- Lindsey Sloat analyzed the data, authored or reviewed drafts of the article, and approved the final draft.
- Tomislav Hengl conceived and designed the experiments, analyzed the data, prepared figures and/or tables, authored or reviewed drafts of the article, and approved the final draft.

## Data Availability

The code is available at GitHub and Zenodo:

- https://github.com/openlandmap/scikit-map/tree/feat_tsirf.

- Leandro Leal Parente, olegsson, Martin Landa, Murat Şahin, RezaM, d-consoli, Carmelo Bonannella, yu-feng-ho, Chris van Diemen, MartijnWitjes, Ondrej Pesek, & Tharles de Sousa Andrade. (2024). openlandmap/scikit-map: Time-Series Iteration-free Reconstruction Framework (TSIRF) v1 (publications). Zenodo. https://doi.org/10.5281/zenodo.14051980.

The benchmark data is available at Zenodo: Consoli, D., Parente, L., Simoes, R., Şahin, M., Tian, X., Witjes, M., & Hengl, T. (2024). Historical time-series reconstruction benchmark dataset of Landsat bi-monthly aggregates from GLAD ARD-2 at 30-m resolution with stratified sampling based on ESA CCI (Version v1) [Data set]. Zenodo. https://doi.org/10.5281/zenodo.11150343.

The 30 m global mosaics in COG is available at GitHub: https://github.com/openlandmap/scikit-map/blob/feat_tsirf/CATALOG.md using the keywords NDVI, SWA, and gap fraction.

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
