# Peer review of "A computational framework for processing time-series of earth observation data based on discrete convolution: global-scale historical Landsat cloud-free aggregates at 30 m spatial resolution"

_PeerJ, doi:10.7717/peerj.18585_

## Round 0.1 · original submission · Major Revisions

I'm glad to let you know that your manuscript received two positive reviews as follows. Though both reviewers think this manuscript has the potential to be published in PeerJ, they also raised quite some issues and concerns. Therefore, major revision is needed.

Reviewer 1 ·

Basic reporting

Uses clear unambiguous English throughout.
Intro shows context.
Literature mostly well referenced.
Figures are relevant and good quality.
Further points:
a) l. 16 remove “extremely”, it is unnecessary
b) l.40 Radeloff et al 2024 Inappropriate reference for content of sentence. Instead cite original Landsat or Sentinel data description publications if any
c) Fig. 1 Fontsize of “m = “ and “n = “ is very small
d) Fig. 2 Improve horizontal & vertical alignment of elements. Unify font size of axis labels.
e) Fig. 6 Legend Marker for Standard LI is wrong
f) L. 841 Montero et al 2023 inappropriate reference for content. Instead cite original references for NDVI, EVI and BSI.
g) Tab 2 does not seem very relevant for this work.
h) Fig. 13b does not seem relevant for this work
i) Fig. 14 the colorbar is very small, consider to enlarge it, also consider to use absolute numbers instead of percentage, to give an indication how spare the timeseries are at each point
j) Fig. 15 illustrates potential future work, I would argue, since you do not actually produce a product fusing Modis and Landsat in this work, it is irrelevant to show this figure. It is enough to mention in one sentence that through clever padding and concatentation of multi-sensor time series, it can be processed jointly with a single kernel to produce a merged product
k) L. 744-760 the discussion of Equi7 and the DGGS, while certainly well written and in general interesting, is irrelevant for this paper. It can be dropped, or at least should be shortened and moved to the future works section as a possible future way to store the data.
l) Moreno-Martinez et. al. (2020) should be cited more prominently in this work, as it appears to me extremely close in both methodology and result. This work (and other similar works that I might be unaware of) should already be mentioned in the introduction. Moreover it should be mentioned in future works, where the fusion of Modis and Landsat is heavily discussed.
m) You could more prominently feature how the availability of the ARCO Landsat dataset will enable downstream machine learning research, such as on vegetation forecasting (https://www.sciencedirect.com/science/article/abs/pii/S003442572030256X , https://openaccess.thecvf.com/content/CVPR2024/html/Benson_Multi-modal_Learning_for_Geospatial_Vegetation_Forecasting_CVPR_2024_paper.html ) or on geospatial foundation models (https://arxiv.org/abs/2310.18660 , https://openaccess.thecvf.com/content/CVPR2024/html/Han_Bridging_Remote_Sensors_with_Multisensor_Geospatial_Foundation_Models_CVPR_2024_paper.html ), which so far often tend to rely on sentinel data, and this dataset allows to go further back in time.
n) L. 142f the sudden introduction of circular convolutions lacks motivation. Latter it appears central for this work, but at first one may wonder why it is necessary to go to periodic extensions. Consider to motivate it briefly, and say why this way of processing is ultimately more computationally efficient, even if first introducing a lot more notation.
o) Clarity of Tab. 1 can be improved: shorten sentences & turn them into bullet points. Potentially also just put an image that represents each kernel.
p) An important chunk of related work to me appears to be leveraging SAR data for gapfilling, since MODIS is mentioned, I believe you should also cite some works that use Sentinel 1.
q) Given you are comparing with fairly simple baselines, I would probably tone down a bit the significance of the 15% improvement in performance.
r) Using sigma as variable for kernel sparsity may be ambiguous given it is usually used to define std. dev.?
s) Figure 9 caption “reluctance”
t) L 487 introduces “OGH Landsat ARCO” – is this the name for the bi-monthly gapfilled Landsat ARCO dataset?
u) I am unsure about the wording in L. 566 – 571. It seems to suggest that commercial cloud providers offer the only potential solution to large-scale processing of remote sensing data. However, (at least in Europe) researcher can relatively easily gain access to compute at HPC centers, and some of those are also close located to data lakes where remote sensing products are stored.
v) L. 571 – 587, the reasoning here is not clear to me. The operations you do seem fairly standard, which is why you may not need a dedicated geospatial library. However you do provide a performant codebase and that is awesome. I do not get why you compare with pyjeo here and then with other libraries in L 615f. There is no need to differentiate yourself so much, just mention you wrote your codebase as a python library that allows to use it for related tasks also. Also mention that compared to existing libraries, your approach is more tailored for the gapfilling tasks and combines performance with easy usability.

Experimental design

Research seems to be within scope of journal.
Research question is well defined.
Investigation is rigorous.
Research is reproducible (open source, detailed description of methodology).

Validity of the findings

I could not check the produced dataset, as the provided STAC endpoint (https://stac.openlandmap.org) did not list it as of June 26. Please clarify in the paper if the derived bi-monthly Landsat dataset is indeed going to be openly accessible. And please provide instruction for how to access it, such that I can assess the validity of the reported findings on the dataset.
Other points:
a) The introduced metric CVRMSE I believe is more commonly known as relative RMSE. To improve clarity I would suggest to rename it in this way. Furthermore, I would suggest to divide not by the Mean of the target time series, but instead by the Std. Dev. of the target time series. The former, which you are doing now, penalizes bands with higher variability, so does not lead to a fair comparison between bands, which appears to be your goal with this metric. The latter would be much fairer in this regard.
b) Please report raw RMSE values in addition to relative RMSE values. To assess the uncertainty of this product for downstream applications, these are even more important.
c) Fig. 7 indicates that the performance of SWAG, especially for the Blue band, is terribly bad, given a relative RMSE of >50%. This indicates it would be good to be more cautious and put more emphasis on reporting the limitations of the bi-monthly gapfilled Landsat dataset. For this, I recommend to include a figure that presents the performance on some artificial Gaps, I.e. mask out the whole images of a few time steps at a few locations and showcase, how the SWAG gap-filling compares to the groundtruth (and possibly to Sen2 also). Make sure to not cherry pick these samples, but instead showcase the breadth of possibilities, e.g. by representing such examples for time series with varying data density and with varying RMSE. Readers should afterwards get a feeling for how SWAG performs on average, how close it can get in the best case, and what some of its failure modes are (and how prevalent these are). Fig. 13a showcasing some artifacts is a good starting point, which I would like to see expanded upon.
d) SWAG is introduced as a new method, however to me it appears to have striking similarity to a Kalman filter. Please discuss this, and either rename & rephrase everything in Kalman filter language (if it is indeed equivalent, or a special case thereof) or highlight the differences to Kalman filters.
e) SIRCLE is introduced as a new method, however it is well known (and in fact leveraged in many libraries), that many operations can be written as convolutions on data. Thus, I wonder if it really necessary to use a new acronym, or if it instead possible to just rely on standard wording, to not confuse the reader.
f) In section 2.2, three different “versions” of SIRCLE are introduced, becoming increasingly more general. To reduce confusion, I suggest rewriting this section, simply explaining Eq. (6) as a convolution, which optionally may be preceded by element-wise scaling and combined with masking.

Additional comments

The paper reads great, and the bi-monthly ARCO Landsat dataset is very relevant and timely. I am assuming you hope for many downstream applications and users to leverage the data, hence I would recommend improving the section on limitations in the paper, just to really make transparent to the user any weaknesses it may have.
To my mind, the biggest contribution of this work is the bi-monthly ARCO Landsat dataset, and not SIRCLE or SWAG. I would encourage you to thus reduce a bit of the framing that you are developing new methodology, and instead try to explain what you did in simple and common terminology, potentially shortening the method section. Briefly cover how the chosen methods allow for processing at the scale necessary for the full landsat archive, and then move to covering the produced dataset in depth. (meaning the correlation plots and the limitation figures I mentioned before).
You may wish to consider renaming SWAG. The acronym is already used in the related field of machine learning, for Stochastic Weight Averaging Gaussian (https://arxiv.org/abs/1902.02476)

Reviewer 2 ·

Basic reporting

This paper generated and distributed the bi-monthly Landsat composite for all the global Landsat archives from the Landsat ARD data produced by University of Maryland and stored in Amazon cloud (1997-2022). In addition, the generation is in the novel SIRCLE framework which represents a significant advancement in the processing of large-scale earth observation time-series data. I like the idea to generalize diverse time-series processing techniques using convolution kernels.

Despite tons of literature published on the algorithm of composition or gap-filling, this paper will stand out as the global dataset has been produced and the data and codes to produce them are published. Despite this, I have a few comments
1, The authors should discuss that one notable disadvantage of the proposed method is the inherent information loss associated with the compositing process. Many traditional Landsat processing algorithms, including classification and change detection, rely on utilizing all available Landsat observations. This utilization of all available Landsat observations has been extended to deep learning algorithms as well, which benefit from the rich temporal information provided by the full suite of observations. In fact, Landsat data itself is often considered insufficient in temporal resolution, which is why Sentinel-2 data are frequently fused to achieve better temporal resolution. By reducing the dataset to composites, there is a risk of losing valuable temporal information that could be crucial for accurately detecting and analyzing changes over time.
The composited data generated by this products should be particularly advantageous for algorithms that rely on spatial context, such as Convolutional Neural Networks (CNNs). These spatially focused algorithms can leverage the high-quality, cloud-free images produced by the compositing process to enhance their performance. However, for applications that depend on the full temporal resolution of Landsat data, such as detailed time-series analysis or change detection requiring all observations, the loss of temporal granularity could be a significant drawback.

2, On the extension of #1, it is important to balance the benefits of reduced data volume and improved spatial quality against the potential disadvantages of information loss in temporal resolution. So what is the logic behind using bi-monthly composites? Why not monthly or bi-weekly? Is this approach related to existing studies and the authors’ examination of the average number of cloud-free Landsat observations globally?


3, A single metric CVRMSE is used. I prefer seeing RMSE also as RMSE is the mostly widely used metric and it has the unit of reflectance (easy to follow) and many other Landsat reconstruction studies such those listed below use RMSE. So with RMSE readers can have an intuitive impression on the algorithm performance in the context of literature.
4, The authors used many equations that contain mistakes or are not well explained. Errors in equations are much more challenging for readers than typos in sentences. I have listed some examples below, and the authors should carefully check all their equations.

Line 78-79, refine the sentence. Currently it reads like cloud mask algorithm should be blamed for gaps. It is cloud not cloud mask algorithm.

Writing and explanation of equations should be very careful as they convey critical information and any mislabel or mistake could confuse readers a lot. However, after checking equations in page 5, I did find quite a few potential mistakes, please check them through the paper. It is better practice to define symbol’s when they first appears, rewrite so that Np Nf are defined. Line 149, should a comma be followed after 2[n]|n<0? Line 150, future past? Line 151, it should be Ns or Nf?

Eq. 5 and Eq. 6 why two different symbol to represent the same Hadamard product in the two equations, respectively? This confuses me a lot and stop me from understanding Eq. (6).

Eq. in 11, what is x(t), does the author mean w(t)?

Can the authors plot the four forms of x(t) using examples and figures (similar to Fig. 2) for readers better understanding? I am lost.

Until later, I realized reconstruction in Section 2.2 follows the bi-monthly composite (please specify this at the beginning of 2.2 if I am correct). Otherwise, it is very difficult to understand as in literature reconstruction can be in many things, for example, a typical example is reconstruction of the full time series, e.g.,

Why 1997 is the start year?

Give citations in Table 1.

After Table 1, I am still lost how Eq. (6) goes? Can we delete Eq. 6 without harming the paper? If so, I strongly suggesting doing so. Less is better.

Line 366, another N, Na. List all the N in a table to explain each of them is a good idea. I lost count of them but I felt it is 8 to 10 different Ns. Hard to follow each of them.

Line 382, so the input GLAD itself is a 16-day composite. Is it better if generated from raw Landsat achieve? I know there is no global ARD now defined at daily resolution with no composite. But this should be discussed and acknowledged.

Line 406, I believe this is the first time I saw wf and it is not defined. Hard to follow.

Lines 430-435 is about how the QA is set for the band. I cannot follow but I felt like this is interpolated from existing QA values. This is not allowed as the original QA is bit-packed and interpolation of bit-packed values are useless.

Again, Wp and Wf???

Fig. 6. CVRMSE and RMSE could both be used.

Fig. 7. CVRMSE and RMSE could both be used.

Line 721, “37% of the pixels are no data, propagating this to the output” This is a very important number – perhaps put this in abstract. And extend this for what is the percentage of pixels with no data for each year

Lin3 738, 8-bit unsigned integers – this is important information as the 12-bit radiometric resolution of Landsat-8 degraded into 8-bit Landsat 5/7. Many users need 12-bit radiometric resolution (e.g., water research) will not use the data by simply reading the abstract.

Line 844-847, a good comment on boundary between deep learning and other data driven approach.

Experimental design

I did not see improper Experimental design. There could be some alternative Experimental designs but they are not compulsory and they are not fair for authors to re-process global data.

Please do pay attention my comment on RMSE which does not global reprocessing.

Validity of the findings

Please do pay attention my comment on RMSE which does not global reprocessing.

Additional comments

NA

---

## Round 0.2 · Major Revisions

Though one reviewer thinks this is ready to be accepted, the other reviewer (R1) raised some issues and some of them are major ones, particularly the access to the global archive to better evaluation of the dataset.

Reviewer 1 ·

Basic reporting

My comments have been addressed.

Experimental design

My comments have been addressed.

Validity of the findings

I have checked the provided files of NDVI for summer and winter 2022 as listed in https://github.com/openlandmap/scikit-map/commit/c021ce1c01ebd307f4cd9cde0609af3611299460#diff-bb476268db5ed6523e97a56f61d3ec99765899bf3269949ee1aac390f48f1b99 .

They seem fine, after quickly looking around I could find two issues, related to edge effects and to striping. Striping is reported in Fig. 15, edge effects around water bodies not yet. See my attached PDF.

Also, how did you normalize NDVI? I assumed the stored range 0-250 represents NDVI 0-1 (because you mask out most water bodies), or does it go for -1 to 1?

While these two COGs are super nice, I am a bit puzzled regarding the writing and framing in the abstract. Do you actually plan to host the 20 TB per band/index as suggested in lines 31-35? Or do you just plan to release these two sample products for 2022?
If the latter, it is necessary to rewrite all parts of the paper that suggest otherwise.
If the former, I am surprised why I have not been able to review these so far, and suggest you share with me the links to the entire catalogue. In particular I would like to review how the product quality changes over time, and if meaningful time series can be derived or if artifacts have to be expected.

Annotated reviews are not available for download in order to protect the identity of reviewers who chose to remain anonymous.

Reviewer 2 ·

Basic reporting

I believed the authors have addressed all of my concerns. Thanks for addressing them properly.

Experimental design

No further comments.

Validity of the findings

No further comments.

Additional comments

No further comments.

---

## Round 0.3 · accepted · Accept

Congratulations and I am looking forward to the final publication!

Reviewer 1 ·

Basic reporting

-

Experimental design

-

Validity of the findings

Thank you for sharing the SWIR band and good luck with acquiring funding for the hosting of the whole dataset.